# Self-Supervised Contrastive Pre-Training for Time Series via Time-Frequency Consistency

**Xiang Zhang**[†] [*]
Harvard University
xiang_zhang@hms.harvard.edu

**Ziyuan Zhao**[*]
Harvard University
ziyuanzhao@college.harvard.edu

**Theodoros Tsiligkaridis**
MIT Lincoln Laboratory
ttsili@ll.mit.edu

**Marinka Zitnik**
Harvard University
marinka@hms.harvard.edu

## Abstract

Pre-training on time series poses a unique challenge due to the potential mismatch between pre-training and target domains, such as shifts in temporal dynamics, fast-evolving trends, and long-range and short-cyclic effects, which can lead to poor downstream performance. While domain adaptation methods can mitigate these shifts, most methods need examples directly from the target domain, making them suboptimal for pre-training. To address this challenge, methods need to accommodate target domains with different temporal dynamics and be capable of doing so without seeing any target examples during pre-training. Relative to other modalities, in time series, we expect that time-based and frequency-based representations of the same example are located close together in the time-frequency space. To this end, we posit that time-frequency consistency (TF-C) — embedding a time-based neighborhood of an example close to its frequency-based neighborhood — is desirable for pre-training. Motivated by TF-C, we define a decomposable pre-training model, where the self-supervised signal is provided by the distance between time and frequency components, each individually trained by contrastive estimation. We evaluate the new method on eight datasets, including electrodiagnostic testing, human activity recognition, mechanical fault detection, and physical status monitoring. Experiments against eight state-of-the-art methods show that TF-C outperforms baselines by 15.4% (F1 score) on average in one-to-one settings (*e.g.*, fine-tuning an EEG-pretrained model on EMG data) and by 8.4% (precision) in challenging one-to-many settings (*e.g.*, fine-tuning an EEG-pretrained model for either hand-gesture recognition or mechanical fault prediction), reflecting the breadth of scenarios that arise in real-world applications. The source code and datasets are available at https://github.com/mims-harvard/TFC-pretraining.

## 1 Introduction

Time series plays important roles in many areas, including clinical diagnosis, traffic analysis, and climate science [1, 2, 3, 4, 5, 6]. While representation learning has considerably advanced analysis of time series [7, 8, 9] more broadly [10], learning generalizable representations for temporal data remains a fundamentally challenging problem [8, 11]. There are numerous immediate benefits from generating such representations, of which pre-training capability is particularly desirable and of

---

[*]These authors contributed equally to this work.

[†]Present address: University of North Carolina at Charlotte, xiang.zhang@uncc.edu

36th Conference on Neural Information Processing Systems (NeurIPS 2022).

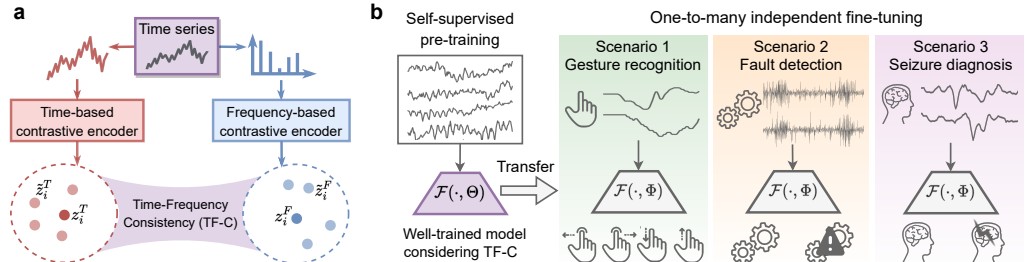

**Figure 1: a.** Illustration of Time-Frequency Consistency (TF-C). Time-based embedding $z_i^{\mathrm{T}}$ and frequency-based embedding $z_i^{\mathrm{F}}$ of time series sample $x_i^{\mathrm{T}}$, along with $\tilde{z}_i^{\mathrm{T}}$ and $\tilde{z}_i^{\mathrm{F}}$ learned from augmentations of $x_i^{\mathrm{T}}$, should be close to each other in the latent time-frequency space. **b.** Leveraging TF-C property in time series to optimize a pre-training model $\mathcal{F}$ with parameters $\Theta$ that get fine-tuned to $\Phi$ on a small scenario-specific dataset.

great practical importance [12, 13]. Central to pre-training is a question of how to process time series in a diverse dataset to greatly improve generalization on new time series coming from different datasets [14, 15, 10]. By training a neural network model on a dataset and transferring it to a new target dataset for fine-tuning, *i.e.*, without explicit retraining on that target data, we expect the resulting performance to be at least as good as that of state-of-the-art models tailored to the target dataset.

However, unfortunately, the expected performance gains are often not realized for a variety of reasons (*e.g.*, distribution shifts, properties of the target dataset unknown during pre-training) [16, 17] that get compounded by the complexity of time series: large variations of temporal dynamics across datasets, varying semantic meaning, irregular sampling, system factors (*e.g.*, different devices or subjects), etc. [18, 17]. This complexity of time series limits the utility of knowledge transfer for pre-training [19, 20]. For example, pre-training a model on a diverse time series dataset with mostly low-frequency components (smooth trends) may not lead to positive transfer on downstream tasks with high-frequency components (transient events) [17]. Examining these challenges can provide clues to what kind of inductive biases could facilitate generalizable representations of time series – this paper offers a strategy for that through a novel time-frequency consistency principle.

In addition, target datasets are not available during pre-training (different from domain adaption [21]; Appendix A), requiring that the pre-training model captures a latent property that holds true for previously unseen target datasets. At the center of this desideratum is the idea of a property that would be shared between pre-training and target datasets and would enable knowledge transfer from pre-training to fine-tuning. In computer vision (CV), pre-training is driven by findings that initial neural layers capture universal visual elements, such as edges and shapes, that are relevant regardless of image style and tasks [22]. In natural language processing (NLP), the foundation for pre-training is given by linguistic principles of semantics and grammar shared across different languages [23]. However, due to the aforementioned temporal complexity, such a principle for pre-training on time series has not yet been established. Moreover, supervised pre-training requires access to large annotated datasets, which limits its use in domains where richly labeled datasets are scarce [24, 25]. For example, in medical applications, labeling data at scale is often infeasible or can be expensive and noisy (experts can disagree on ground-truth labeling [26, 27], *e.g.*, whether an ECG signal indicates a normal vs. abnormal rhythm) [28, 29]. To mitigate these issues, self-supervised learning emerged as a promising strategy to sidestep the lack of labeled datasets [30].

**Present work.** We introduce a strategy for self-supervised pre-training in time series by modeling Time-Frequency Consistency (TF-C). TF-C specifies that time-based and frequency-based representations, learned from the same time series sample, should be closer to each other in the time-frequency space than representations of different time series samples. Specifically, we adopt contrastive learning in time-space to generate a time-based representation. In parallel, we propose a set of novel augmentations based on the characteristic of the frequency spectrum and produce a frequency-based embedding through contrastive instance discrimination. This is the first work to develop frequency-based contrastive augmentation to leverage rich spectral information and explore time-frequency consistency in time series. The pre-training objective is to minimize the distance between time-based and frequency-based embeddings using a novel consistency loss (Figure 1 (a)). The self-supervised loss is used to optimize the pre-training model and enforce consistency between time and frequency domains in the latent space. The learned relationship encoded in model parameters are transferred to initialize the fine-tuning model and improve performance in datasets of interest (Figure 1 (b)).

We evaluate the TF-C model on eight time series datasets under two evaluation settings (*i.e.*, one-to-one and one-to-many). The eight datasets cover a large set of variations: different numbers of channels (from univariate to 9-channel multivariate), varying time series lengths (from 128 to 5,120), different sampling rates (from 16 Hz to 4,000 Hz), different scenarios (neurological healthcare, human activity recognition, mechanical fault detection, physical status monitoring, etc.) and diverse types of signals (EEG, EMG, ECG, acceleration, and vibration). We compare TF-C approach to eight state-of-the-art baselines. Results show that TF-C achieves positive transfer, outperforming all baselines by a large margin of 15.4% (F1 score) on average. Further, the approach outperforms the strongest baselines with an improvement of up to 7.2% in the F1 score. Finally, the TF-C approach improves prior work by 8.4% in precision (when pre-training the model on sleep EEG signals and fine-tuning it on hand-gesture recognition) in challenging one-to-many setups that apply the same pre-trained model to multiple independent fine-tuning datasets.

## 2 Related Work

**Pre-training for time series.** Although there are studies on self-supervised representation learning for time series [7, 8, 31, 32] and self-supervised pre-training for images [33, 34, 35, 24], the intersection of these two areas, *i.e.*, self-supervised pre-training for time series, remains underexplored. In time series, it's not obvious what reasonable assumptions can bridge pre-training and target datasets. Hence, pre-training models in CV [36, 37, 14] and NLP [10, 15, 38] are not directly applicable due to data modality mismatch, and the existing results leave room for improvement [31, 39, 40]. Shi *et al.* [12] developed the only model to date that is explicitly designed for self-supervised time series pre-training. The model captures the local and global temporal pattern, but it is not convincing why the designed pretext task can capture generalizable representations. Although several studies applied transfer learning in the context of time series [7, 8, 18, 41], there is no foundation yet of which conceptual properties are most suitable for pre-training on time series and why. Addressing this gap, we show that TF-C, designed to be invariant to different time-series datasets, can produce generalizable pre-training models.

Unlike domain adaptation [21, 42] that requires access to target datasets during training, pre-training models do not have access to fine-tuning datasets. As a result, one needs to identify a generalizable time-series property to benefit from pre-training. Further, self-supervised domain adaptation does not need labels in the target dataset but still requires labels for model training [43, 44]. In contrast, TF-C does not need any labels during pre-training.

**Contrastive learning with time series.** Contrastive learning, a popular type of self-supervised learning, aims to learn an encoder that maps inputs into an embedding space such that positive sample pairs (original augmentation and another alternative augmentation/view of the same input sample) are pulled closer and negative sample pairs (original augmentation and an alternative input sample augmentation) are pushed apart [30, 45]. Contrastive learning in time series is less investigated in comparison, partly due to the challenge of identifying augmentations that capture key invariance properties in time series data. For example, CLOCS defines adjacent time segments as positive pairs [41], and TNC assumes overlapping temporal neighborhoods have similar representations [46]. These methods leverage temporal invariance to define positive pairs which are used to calculate contrastive loss, but other invariances, such as transformation invariance (*e.g.*, SimCLR [40]), contextual invariance (*e.g.*, TS2vec [47] and TS-TCC [48]) and augmentations are possible. In this work, we propose an augmentation bank that exploits multiple invariances to generate diverse augmentations (Sec. 4.1), which adds richness to the pre-training model [48]. Importantly, we propose frequency-based augmentations by perturbing the frequency spectrum of time series (*e.g.*, adding or removing the frequency components and manipulating their amplitude; more details in Sec. 4.2) to learn better representations by exposing the model to a local range of frequency variations. In previous work, CoST processes sequential signals through the frequency domain, but the augmentations are still implemented in time space [49]. Similarly, although BTSF [50] involves frequency domain, its data transformation is solely implemented in the time domain using instance-level dropout. Additional commentary on differences between CoST and BTSF is in Appendix B. To the best of our knowledge, this is the first work that directly perturbs the frequency spectrum to leverage frequency-invariance for contrastive learning. Further, we develop a pre-training model that subjects to TF-C upon two individual contrastive encoders.

# 3 Problem Formulation

We are given a pre-training dataset $\mathcal{D}^{\text{pret}} = \{ \boldsymbol{x}_i^{\text{pret}} \mid i = 1, \dots, N \}$ of unlabeled time series samples where sample $\boldsymbol{x}_i^{\text{pret}}$ has $K^{\text{pret}}$ channels and $L^{\text{pret}}$ timestamps. Let $\mathcal{D}^{\text{tune}} = \{ (\boldsymbol{x}_i^{\text{tune}}, y_i) \mid i = 1, \dots, M \}$ be a fine-tuning (*i.e.*, target; target and fine-tuning are used interchangeably) dataset of labeled time series samples, each having $K^{\text{tune}}$ channels and $L^{\text{tune}}$ timestamps. Furthermore, every sample $\boldsymbol{x}_i^{\text{tune}}$ is associated with a label $y_i \in \{1, \dots, C\}$, where $C$ is the number of classes. Without loss of generality, in the following descriptions, we focus on univariate (single-channel) time series, while noting that our approach can accommodate multivariate time series of varying lengths across datasets (shown in experiments in Sec. 5.2). We use superscript symbol $\sim$ to denote contrastive augmentations. We note that $\boldsymbol{x}_i^{\text{T}} \equiv \boldsymbol{x}_i$ denotes an input time series sample, and $\boldsymbol{x}_i^{\text{F}}$ denotes discrete frequency spectrum of $\boldsymbol{x}_i$.

**Problem (Self-Supervised Contrastive Pre-Training For Time Series).** *Given are an unlabeled pre-training dataset $\mathcal{D}^{pret}$ with $N$ samples and a target dataset $\mathcal{D}^{tune}$ with $M$ samples ($M \ll N$). The goal is to use $\mathcal{D}^{pret}$ to pre-train a model $\mathcal{F}$ so that by fine-tuning model parameters on $\mathcal{D}^{tune}$, the fine-tuned model produces generalizable representations $\boldsymbol{z}_i^{tune} = \mathcal{F}(\boldsymbol{x}_i^{tune})$ for every $\boldsymbol{x}_i^{tune}$.*

We follow an established setup, *e.g.*, [41]: for pre-training, only the unlabeled dataset $\mathcal{D}^{\text{pret}}$ is available while, for fine-tuning, a small labeled dataset $\mathcal{D}^{\text{tune}}$ can be used. In short, a model $\mathcal{F}$ is pre-trained on the unlabeled time series dataset $\mathcal{D}^{\text{pret}}$ and its optimized model parameters $\Theta$ are fine-tuned to go from $\mathcal{F}(\cdot, \Theta)$ to $\mathcal{F}(\cdot, \Phi)$ using the dataset $\mathcal{D}^{\text{tune}}$. The $\Phi$ denotes fine-tuned model parameters. Note that this problem (*i.e.*, $\mathcal{D}^{\text{pret}}$ is independent of the target dataset) is distinct from domain adaptation as fine-tuning dataset $\mathcal{D}^{\text{tune}}$ is not accessed during pre-training. As a result, the pre-trained model can be used with many different fine-tuning datasets without re-training.

**Rationale for Time-Frequency Consistency (TF-C).** The central idea is to identify a general property that is preserved across time series datasets and use it to induce transfer learning for effective pre-training. The time domain shows how sensor readouts change with time, whereas the frequency domain shows how much of the signal lies within each frequency component over the entire spectrum [51]. Explicitly considering the frequency domain can provide an understanding of time series behavior that cannot be directly captured solely in the time domain [52]. However, existing contrastive methods (*e.g.*, [47, 48]) focus exclusively on modeling the time domain and ignore the frequency domain altogether. One can argue that approach is sufficient in the case of high-capacity methods as time and frequency domains are different views of the same data [53], which can be cross-translated using transformation, such as Fourier and inverse Fourier [54, 52]. The relationship between the two domains, grounded in signal processing theory, provides an invariance that is valid regardless of the time series distribution [55, 56] and thus serve as an inductive bias for pre-training. Appendix C provides a commentary with analogies for images. Approaching this invariance through the lens of representation learning, we next formulate Time-Frequency Consistency (TF-C). The TF-C property postulates there exists a latent time-frequency space such that for every sample $\boldsymbol{x}_i$, time-based representation $\boldsymbol{z}_i^{\text{T}}$ and frequency-based representation $\boldsymbol{z}_i^{\text{F}}$ of the same sample, together with their local augmentations (defined later), are close to each other in the latent space.

**Representational Time-Frequency Consistency (TF-C).** *Let $\boldsymbol{x}_i$ be a time series and $\mathcal{F}$ be a model satisfying TF-C. Then, time-based representation $\boldsymbol{z}_i^{T}$ and frequency-based representation $\boldsymbol{z}_i^{F}$ as well as representations of $\boldsymbol{x}_i$'s local augmentations are proximal in the latent time-frequency space.*

Our strategy is to use dataset $\mathcal{D}^{\text{pret}}$ to induce TF-C in $\mathcal{F}$'s model parameters $\Theta$, which, in turn, are used to initialize the target model on $\mathcal{D}^{\text{tune}}$ and produce generalizable representations for downstream prediction. The invariant nature of TF-C means that the approach can bridge $\mathcal{D}^{\text{pret}}$ and $\mathcal{D}^{\text{tune}}$ even when large discrepancies exist between them (in terms of temporal dynamics, semantic meaning, etc.), providing a vehicle for a general pre-training on time series.

To realize TF-C, our model $\mathcal{F}$ has four components (Figure 2): a time encoder $G_{\text{T}}$, a frequency encoder $G_{\text{F}}$, and two cross-space projectors $R_{\text{T}}$ and $R_{\text{F}}$ that map time-based and frequency-based representations, respectively, to the same time-frequency space. Together, the four components provide a way to embed $\boldsymbol{x}_i$ to the latent time-frequency space such that the time-based embedding $\boldsymbol{z}_i^{\text{T}} = R_{\text{T}}(G_{\text{T}}(\boldsymbol{x}_i^{\text{T}}))$ and the frequency-based embedding $\boldsymbol{z}_i^{\text{F}} = R_{\text{F}}(G_{\text{F}}(\boldsymbol{x}_i^{\text{F}}))$ are close together.

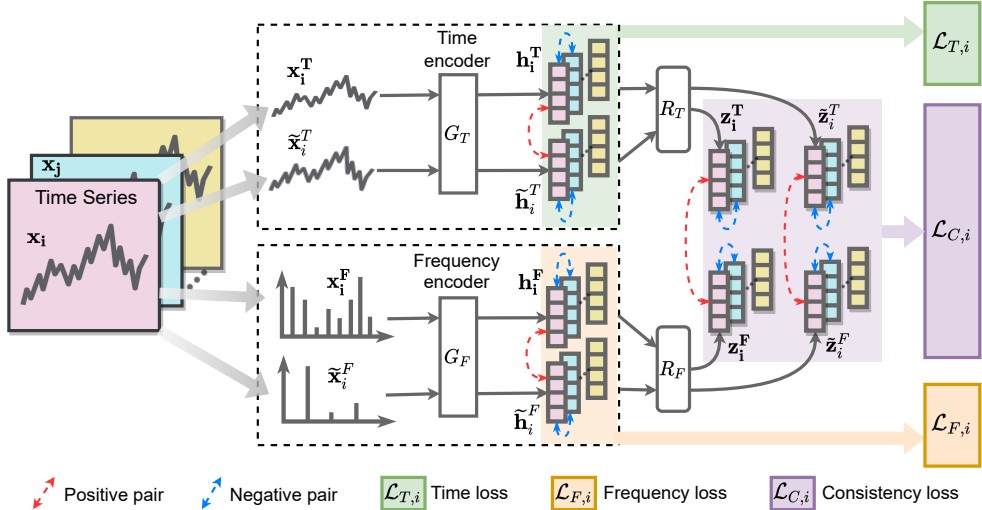

**Figure 2: Overview of TF-C approach.** Our TF-C pre-training model $\mathcal{F}$ has four components: a time encoder $G_\text{T}$, a frequency encoder $G_\text{F}$, and two cross-space projectors $R_\text{T}$ and $R_\text{F}$. For an input time series $\boldsymbol{x}_i$, the model produces time-based representations (*i.e.*, $\boldsymbol{z}_i^\text{T}$ and $\widetilde{\boldsymbol{z}}_i^\text{T}$ of input $\boldsymbol{x}_i$ and its augmented version, respectively) and frequency-based representations (*i.e.*, $\boldsymbol{z}_i^\text{F}$ and $\widetilde{\boldsymbol{z}}_i^\text{F}$ of input $\boldsymbol{x}_i$ and its augmented version, respectively). The TF-C property is realized by promoting the alignment of time- and frequency-based representations in the latent time-frequency space, providing a vehicle for transferring $\mathcal{F}$ to a target dataset not seen before.

## 4 Our Approach

Next, we present the architecture of the developed self-supervised contrastive pre-training model $\mathcal{F}$. Unless specified otherwise, the data mentioned in this section are from pre-training dataset and the superscript $^\text{pret}$ is omitted for simplification. Here we describe the model using univariate time series as an example, but our model can be straightforwardly applied to multivariate time series (Sec 5).

### 4.1 Time-based Contrastive Encoder

For a given input time series sample $\boldsymbol{x}_i$, we generate an augmentation set $\mathcal{X}_i^\text{T}$ through a time-based augmentation bank $\mathcal{B}^\text{T} : \boldsymbol{x}_i^\text{T} \to \mathcal{X}_i^\text{T}$. Each element $\widetilde{\boldsymbol{x}}_i^\text{T} \in \mathcal{X}_i^\text{T}$ is augmented from $\boldsymbol{x}_i$ based on the temporal characteristics. Here, the time-based augmentation bank includes jittering, scaling, time-shifts, and neighborhood segments, all well-established in contrastive learning [40, 48, 41]. We develop an augmentation bank to produce diverse augmentations (rather than a single type of augmentation) and expose the model to complex temporal dynamics, which produces more robust time-based embeddings [48].

For the input $\boldsymbol{x}_i$, we randomly select an augmented sample $\widetilde{\boldsymbol{x}}_i^\text{T} \in \mathcal{X}_i^\text{T}$ and feed into a contrastive time encoder $G_\text{T}$ that maps samples to embeddings. We have $\boldsymbol{h}_i^\text{T} = G_\text{T}(\boldsymbol{x}_i^\text{T})$ and $\widetilde{\boldsymbol{h}}_i^\text{T} = G_\text{T}(\widetilde{\boldsymbol{x}}_i^\text{T})$. As $\widetilde{\boldsymbol{x}}_i^\text{T}$ is generated based on $\boldsymbol{x}_i^\text{T}$, after passing through $G_\text{T}$, we assume the embedding of $\boldsymbol{x}_i^\text{T}$ is close to the embedding of $\widetilde{\boldsymbol{x}}_i^\text{T}$ but far away from the embedding of $\boldsymbol{x}_j^\text{T}$ and $\widetilde{\boldsymbol{x}}_j^\text{T}$ that are derived from another sample $\boldsymbol{x}_j^\text{T} \in \mathcal{D}^\text{pret}$ [34, 47, 41]. In specific, we select the positive pair as $(\boldsymbol{x}_i^\text{T}, \widetilde{\boldsymbol{x}}_i^\text{T})$ and negative pairs as $(\boldsymbol{x}_i^\text{T}, \boldsymbol{x}_j^\text{T})$ and $(\boldsymbol{x}_i^\text{T}, \widetilde{\boldsymbol{x}}_j^\text{T})$ [34].

**Contrastive time loss.** To maximize the similarity within a positive pair and minimize the similarity within a negative pair, we adopt the NT-Xent (the normalized temperature-scaled cross entropy loss) as distance function $d$ which is widely used in contrastive learning [34, 40]. In specific, we define the loss function of the time-based contrastive encoder in terms of sample $\boldsymbol{x}_i^\text{T}$ as:

$$\mathcal{L}_{\text{T},i} = d(\boldsymbol{h}_i^\text{T}, \widetilde{\boldsymbol{h}}_i^\text{T}, \mathcal{D}^\text{pret}) = -\log \frac{\exp(\text{sim}(\boldsymbol{h}_i^\text{T}, \widetilde{\boldsymbol{h}}_i^\text{T})/\tau)}{\sum_{\boldsymbol{x}_j \in \mathcal{D}^\text{pret}} \mathbb{1}_{i \neq j} \exp(\text{sim}(\boldsymbol{h}_i^\text{T}, G_\text{T}(\boldsymbol{x}_j))/\tau)}, \qquad (1)$$

where $\text{sim}(\boldsymbol{u}, \boldsymbol{v}) = \boldsymbol{u}^T \boldsymbol{v} / \|\boldsymbol{u}\| \|\boldsymbol{v}\|$ denotes the cosine similarity, the $\mathbb{1}_{i \neq j}$ is an indicator function that equals to 0 when $i = j$ and 1 otherwise, and $\tau$ is a temporal parameter to adjust scale. The $\boldsymbol{x}_j \in \mathcal{D}^\text{pret}$ refers to a different time series sample or its augmented sample. This loss function

urges the time encoder $G_\text{T}$ to generate closer time-based embeddings for positive pairs and push the embeddings for negative pairs apart from each other.

## 4.2 Frequency-based Contrastive Encoder

We generate the frequency spectrum $\boldsymbol{x}_i^\text{F}$ from a time series sample $\boldsymbol{x}_i^\text{T}$ through a transform operator (*e.g.*, Fourier Transformation [54]). The frequency information in time series is universal and plays a key role in classic signal processing [57, 53, 55], but it is rarely investigated in self-supervised contrastive representation learning for time series [58]. In this section, we develop augmentation method to perturb $\boldsymbol{x}_i^\text{F}$ based on characteristics of frequency spectra and show how to generate frequency-based representations.

As every frequency component in the frequency spectrum denotes a basis function (*e.g.*, sinusoidal function for Fourier transformation) with the corresponding frequency and amplitude, we perturb the frequency spectrum by adding or removing frequency components. A small perturbation in the frequency domain may cause large changes to the temporal patterns in the time domain [55]. To make sure the perturbed time series is still similar to the original sample (not only in frequency domain but also in time domain; Figure 6), we use a small budget $E$ in the perturbations where $E$ denotes the number of frequency components we manipulate. While removing frequency components, we randomly select $E$ frequency components and set their amplitudes to $0$. While adding frequency components, we randomly choose $E$ frequency components from the ones have smaller amplitude than $\alpha \cdot A_m$, and increase their amplitude to $\alpha \cdot A_m$. The $A_m$ is the maximum amplitude in the frequency spectrum and $\alpha$ is a pre-defined coefficient to adjust the scale of the perturbed frequency component ($\alpha = 0.5$ in this work). We produce an augmentation set $\mathcal{X}_i^\text{F}$ for $\boldsymbol{x}_i^\text{F}$ through frequency-augmentation bank $\mathcal{B}^\text{F} : \boldsymbol{x}_i^\text{F} \to \mathcal{X}_i^\text{F}$. As described above, we have two augmentation methods (*i.e.*, removing or adding frequency components) in $\mathcal{B}^\text{F}$, $|\mathcal{X}_i^\text{F}| = 2$. Details on the exploration of frequency augmentation strategies are covered in Appendix J.

We utilize a frequency encoder $G_\text{F}$ to map the frequency spectrum (*e.g.*, $\boldsymbol{x}_i^\text{F}$) to a frequency-based embedding (*e.g.*, $\boldsymbol{h}_i^\text{F} = G_\text{F}(\boldsymbol{x}_i^\text{F})$). We assume the frequency encoder $G_\text{F}$ can learn similar embedding for the original frequency spectrum $\boldsymbol{x}_i^\text{F}$ and a slightly perturbed frequency spectrum $\widetilde{\boldsymbol{x}}_i^\text{F} \in \mathcal{X}_i^\text{F}$. Thus, we set the positive pair as $(\boldsymbol{x}_i^\text{F}, \widetilde{\boldsymbol{x}}_i^\text{F})$ and the negative pairs as $(\boldsymbol{x}_i^\text{F}, \boldsymbol{x}_j^\text{F})$ and $(\boldsymbol{x}_i^\text{F}, \widetilde{\boldsymbol{x}}_j^\text{F})$.

**Contrastive frequency loss.** We calculate frequency-based contrastive loss for sample $\boldsymbol{x}_i$ as:

$$\mathcal{L}_{\text{F},i} = d(\boldsymbol{h}_i^\text{F}, \widetilde{\boldsymbol{h}}_i^\text{F}, \mathcal{D}^\text{pret}) = -\log \frac{\exp(\text{sim}(\boldsymbol{h}_i^\text{F}, \widetilde{\boldsymbol{h}}_i^\text{F})/\tau)}{\sum_{\boldsymbol{x}_j \in \mathcal{D}^\text{pret}} \mathbb{1}_{i \neq j} \exp(\text{sim}(\boldsymbol{h}_i^\text{F}, G_\text{F}(\boldsymbol{x}_j))/\tau)}. \tag{2}$$

In preliminary experiments, we find that the value of $\tau$ has little effect on performance and use the same $\tau$ throughout all experiments. The $\mathcal{L}_{\text{F},i}$ yield a frequency encoder $G_\text{F}$ producing embeddings invariant to frequency spectrum perturbations.

## 4.3 Time-Frequency Consistency

We develop a consistency loss item $\mathcal{L}_{\text{C},i}$ to urge the learned embeddings to satisfy TF-C: for a given sample, its time-based and frequency-based embeddings (and their local neighborhoods) are supposed to be close to each other (see Sec. 3 for justification). To make sure the distance between embeddings is measurable, we map $\boldsymbol{h}_i^\text{T}$ from time space and $\boldsymbol{h}_i^\text{F}$ from frequency space to a joint time-frequency space through projectors $R_\text{T}$ and $R_\text{F}$, respectively. In specific, for every input sample $\boldsymbol{x}_i$, we have four embeddings, which are $\boldsymbol{z}_i^\text{T} = R_\text{T}(\boldsymbol{h}_i^\text{T})$, $\widetilde{\boldsymbol{z}}_i^\text{T} = R_\text{T}(\widetilde{\boldsymbol{h}}_i^\text{T})$, $\boldsymbol{z}_i^\text{F} = R_\text{F}(\boldsymbol{h}_i^\text{F})$, and $\widetilde{\boldsymbol{z}}_i^\text{F} = R_\text{F}(\widetilde{\boldsymbol{h}}_i^\text{F})$. The first two embeddings are generated based on temporal characteristics and the latter two embeddings are produced based on the properties of frequency spectrum.

To enforce the embeddings in the time-frequency space subject to TF-C, we design a consistency loss $\mathcal{L}_{\text{C},i}$ that measures the distance between a time-based embedding and a frequency-based embedding. We use $S_i^\text{TF} = d(\boldsymbol{z}_i^\text{T}, \boldsymbol{z}_i^\text{F}, \mathcal{D}^\text{pret})$ to denote the distance between $\boldsymbol{z}_i^\text{T}$ and $\boldsymbol{z}_i^\text{F}$). Similarly, we define $S_i^{\widetilde{\text{T}}\text{F}}$, $S_i^{\text{T}\widetilde{\text{F}}}$, and $S_i^{\widetilde{\text{T}}\widetilde{\text{F}}}$. Note, in this time-frequency space, we don't consider the distance between $\boldsymbol{z}_i^\text{T}$ and $\widetilde{\boldsymbol{z}}_i^\text{T}$ where the two embeddings are from the same domain (*i.e.*, time domain). The same applies to pair the distance between $\boldsymbol{z}_i^\text{F}$ and $\widetilde{\boldsymbol{z}}_i^\text{F}$. We have already considered information of above two pairs in the calculation of $\mathcal{L}_{\text{T},i}$ and $\mathcal{L}_{\text{F},i}$.

Next, let's closely observe $S_i^{\text{TF}}$ and $S_i^{\widetilde{\text{TF}}}$ that involve three embeddings: $z_i^{\text{T}}$, $z_i^{\text{F}}$, and $\widetilde{z}_i^{\text{F}}$. Here, $z_i^{\text{T}}$ and $z_i^{\text{F}}$ are learned from the original sample ($x_i^{\text{T}}$ and $x_i^{\text{F}}$) while $\widetilde{z}_i^{\text{F}}$ is learned from the augmented $\widetilde{x}_i^{\text{F}}$. Thus, intuitively, $z_i^{\text{T}}$ should be closer to $z_i^{\text{F}}$ in comparison to $\widetilde{z}_i^{\text{F}}$. Motivated by the relative relationship, we encourage the proposed model to learn a $S_i^{\text{TF}}$ that is smaller than $S_i^{\widetilde{\text{TF}}}$. Inspired by the triplet loss [59], we design $(S_i^{\text{TF}} - S_i^{\widetilde{\text{TF}}} + \delta)$ as a term of consistency loss $\mathcal{L}_{\text{C},i}$ where $\delta$ is a given constant margin to keep negative samples far apart [60]. This term optimizes the model towards a smaller $S_i^{\text{TF}}$ and relatively larger $S_i^{\widetilde{\text{TF}}}$. Similarly, $S_i^{\text{TF}}$ is supposed to be smaller than $S_i^{\widetilde{\text{TF}}}$ and $S_i^{\widetilde{\text{TF}}}$. In summary, we calculate the consistency loss $\mathcal{L}_{\text{C},i}$ for sample $x_i$ by:

$$\mathcal{L}_{\text{C},i} = \sum_{S^{\text{pair}}} (S_i^{\text{TF}} - S_i^{\text{pair}} + \delta), \quad S^{\text{pair}} \in \{S_i^{\widetilde{\text{TF}}}, S_i^{\widetilde{\text{TF}}}, S_i^{\widetilde{\text{TF}}}\}, \tag{3}$$

where $S_i^{\text{pair}}$ denotes the distance between a time-based embedding (*e.g.*, $z_i^{\text{T}}$ or $\widetilde{z}_i^{\text{T}}$) and a frequency-based embedding (*e.g.*, $z_i^{\text{F}}$ or $\widetilde{z}_i^{\text{F}}$). In each pair, there is at least one embedding that is derived from augmented sample instead of the original sample. The $\delta$ is a pre-defined constant. By combining all the triplet loss items, $\mathcal{L}_{\text{C}}$ encourages the pre-training model to capture the consistency between time-based and frequency-based embeddings in model optimization. Note, although the Eq. 3 does not explicitly measure the loss across different time series samples (*e.g.*, $x_i$ and $x_j$), the cross-sample relationships are implicitly covered in the calculation of $S_i^{\text{TF}}$ and $S_i^{\text{pair}}$.

## 4.4 Implementation and Technical Details

The overall loss function in pre-training has three terms. First, the time-based contrastive loss $\mathcal{L}_{\text{T}}$ urges the model to learn embeddings invariant to temporal augmentations. Second, the frequency-based contrastive loss $\mathcal{L}_{\text{F}}$ promotes learning of embeddings invariant to frequency spectrum-based augmentations. Third, the consistency loss $\mathcal{L}_{\text{C}}$ guides the model to retain the consistency between time-based and frequency-based embeddings. In summary, the pre-training loss is defined as:

$$\mathcal{L}_{\text{TF-C},i} = \lambda(\mathcal{L}_{\text{T},i} + \mathcal{L}_{\text{F},i}) + (1 - \lambda)\mathcal{L}_{\text{C},i} \tag{4}$$

where $\lambda$ controls the relative importance of the contrastive and consistency losses. We calculate the total loss by summing $\mathcal{L}_{\text{TF-C},i}$ across all pre-training samples. In implementation, the contrastive losses are calculated within the batch. From our problem definition, the model $\mathcal{F}$ we want to learn is the combination of neural networks $G_{\text{T}}$, $R_{\text{T}}$, $G_{\text{F}}$, and $R_{\text{F}}$. When pre-training is completed, we store parameters of entire model, and denote it as $\mathcal{F}(\cdot, \Theta)$ where $\Theta$ represents all trainable parameters. When a sample $x_i^{\text{tune}}$ is presented, fine-tuned model $\mathcal{F}$ generates an embedding $z_i^{\text{tune}}$ via concatenation as: $z_i^{\text{tune}} = \mathcal{F}(x_i^{\text{tune}}, \Phi) = [z_i^{\text{tune,T}}; z_i^{\text{tune,F}}]$ where $\Phi$ are fine-tuned model's parameters.

## 5 Experiments

We compare the developed TF-C model with 10 baselines on 8 diverse datasets. We investigate the time series classification tasks in the context of one-to-one and one-to-many transfer learning setups (the many-to-one setting is fundamentally different as discussed in Appendix K). We also assess TF-C in extensive downstream tasks including clustering and anomaly detection.

**Datasets.** (1) **SLEEPEEG** [61] has 371,055 univariate brainwaves (EEG; 100 Hz) collected from 197 individuals. Each sample is associated with one of five sleeping stages. (2) **EPILEPSY** [62] monitors the brain activities of 500 subjects with single-channel EEG sensor (174 Hz). A sample is labeled in binary based on whether the subject has epilepsy or not. (3) **FD-A** [63] gathers the vibration signals from rolling bearing from a mechanical system aiming at fault detection. Every sample has 5,120 timestamps and an indicator for one out of three mechanical device states. (4) **FD-B** [63] has the same setting as the **FD-A** but the rolling bearings are performed in different working conditions (*e.g.*, varying rotational speed). (5) **HAR** [64] has 10,299 9-dimension samples from 6 daily activities. (6) **GESTURE** [65] includes 440 samples that are collected from 8 hand gestures recorded by an accelerometer. (7) **ECG** [26] contains 8,528 single-sensor ECG recordings with sorted into four classes based on human physiology. (8) **EMG** [66] consists of 163 EMG samples with 3-class labels implying muscular diseases. Dataset labels are not used in pre-training. Further dataset statistics are in Appendix D and Table 3.

**Baselines.** We consider 10 baseline methods. This includes 8 state-of-the-art methods: TS-SD [12], TS2vec [47], CLOCS [41], Mixing-up [18], TS-TCC [48], SimCLR [40], TNC [46], and CPC [30].

**Table 1: One-to-one pre-training evaluation (Scenario 3).** Pre-training is performed on HAR, followed by fine-tuning on GESTURE. Results for other three scenarios are shown in Tables 4-6.

| Models | Accuracy | Precision | Recall | F1 score | AUROC | AUPRC |
|---|---|---|---|---|---|---|
| **Non-DL (KNN)** | $0.6766_{\pm 0.0000}$ | $0.6500_{\pm 0.0000}$ | $0.6821_{\pm 0.0000}$ | $0.6442_{\pm 0.0000}$ | $0.8190_{\pm 0.0000}$ | $0.5231_{\pm 0.0000}$ |
| **Random Init.** | $0.4219_{\pm 0.0865}$ | $0.4751_{\pm 0.0925}$ | $0.4963_{\pm 0.1026}$ | $0.4886_{\pm 0.0967}$ | $0.7129_{\pm 0.1206}$ | $0.3358_{\pm 0.1194}$ |
| **TS-SD** | $0.6937_{\pm 0.0533}$ | $0.6806_{\pm 0.0496}$ | $0.6883_{\pm 0.0525}$ | $0.6785_{\pm 0.0495}$ | $0.8708_{\pm 0.0305}$ | $0.6261_{\pm 0.0790}$ |
| **TS2vec** | $0.6453_{\pm 0.0260}$ | $0.6287_{\pm 0.0339}$ | $0.6451_{\pm 0.0218}$ | $0.6261_{\pm 0.0294}$ | $0.8890_{\pm 0.0054}$ | $0.6670_{\pm 0.0118}$ |
| **CLOCS** | $0.4731_{\pm 0.0229}$ | $0.4639_{\pm 0.0432}$ | $0.4766_{\pm 0.0266}$ | $0.4392_{\pm 0.0198}$ | $0.8161_{\pm 0.0068}$ | $0.4916_{\pm 0.0103}$ |
| **Mixing-up** | $0.7183_{\pm 0.0123}$ | $0.7001_{\pm 0.0166}$ | $0.7183_{\pm 0.0123}$ | $0.6991_{\pm 0.0145}$ | $\mathbf{0.9127_{\pm 0.0018}}$ | $0.7654_{\pm 0.0071}$ |
| **TS-TCC** | $0.7593_{\pm 0.0242}$ | $0.7668_{\pm 0.0257}$ | $0.7566_{\pm 0.0231}$ | $0.7457_{\pm 0.0210}$ | $0.8866_{\pm 0.0040}$ | $0.7217_{\pm 0.0121}$ |
| **SimCLR** | $0.4383_{\pm 0.0652}$ | $0.4255_{\pm 0.1072}$ | $0.4383_{\pm 0.0652}$ | $0.3713_{\pm 0.0919}$ | $0.7721_{\pm 0.0559}$ | $0.4116_{\pm 0.0971}$ |
| **TF-C (Ours)** | $\mathbf{0.7824_{\pm 0.0237}}$ | $\mathbf{0.7982_{\pm 0.0496}}$ | $\mathbf{0.8011_{\pm 0.0322}}$ | $\mathbf{0.7991_{\pm 0.0296}}$ | $0.9052_{\pm 0.0136}$ | $\mathbf{0.7861_{\pm 0.0149}}$ |

The TS2Vec, TS-TCC, SimCLR, TNC, and CPC are designed for representation learning on a single dataset rather than for transfer learning, so we apply them to fit our settings and make the results comparable. As the training of TNC and CPC are very time-consuming and relatively less competitive (Table 4), we only compare them in the one-to-one setting (scenario 1) while not in other experiments. To examine the utility of pre-training, we consider two additional approaches that are applied directly to fine-tuning datasets without any pre-training: Non-DL (a non-deep learning KNN model) and Random Init. (randomly initializes the fine-tuning model). The evaluation metrics are accuracy, precision (macro-averaged), recall, F1 score, AUROC, and AUPRC.

**Implementation.** We use two 3-layer 1-D ResNets [67] as backbones for encoders $G_T$ and $G_F$. Our datasets contain long time series (samples in FD-A and FD-B have 5,120 observations), and preliminary experiments identified ResNet as a better option than a Transformer variant [68]. We use 2 fully-connected layers for $R_T$ and $R_F$, with no sharing of parameters. We set $E = 1$ and $\alpha = 0.5$ in frequency augmentations and $\tau = 0.2$, $\delta = 1$, $\lambda = 0.5$ in loss functions. Reported are mean and standard deviation values across 5 independent runs (both pre-training and fine-tuning) on the same data split. Results for KNN (K=2) do not change so the standard deviation is zero. Method details and hyper-parameter selection are in Appendix E.

## 5.1 Results: One-to-One Pre-Training Evaluation

**Setup.** In one-to-one evaluation, we pre-train a model on *one* pre-training dataset and use it for fine-tuning on *one* target dataset only. Scenario 1 (SLEEPEEG → EPILEPSY): Pre-training is done on SLEEPEEG and fine-tuning on EPILEPSY. While both datasets describe a single-channel EEG, the signals are from different channels/positions on scalps, track different physiology (sleep vs. epilepsy), and are collected from different patients. Scenario 2 (FD-A → FD-B): Datasets describe mechanical devices that operate in different working conditions, including rotational speed, load torque, and radial force. Scenario 3 (HAR → GESTURE): Datasets record different activities (6 types of human daily activities vs. 8 hand gestures). While both datasets contain acceleration signals, HAR has 9 channels while GESTURE has 1 channel. Scenario 4 (ECG → EMG): While both are physiological datasets, the ECG records the electrical signal from the heart whereas EMG measures muscle response in response to a nerve's stimulation of the muscle. We note that the discrepancies between pre-training and fine-tuning datasets in the above four scenarios are substantial, and they cover a diverse range of variation in time series datasets: varying semantic meaning, sampling frequency, time series length, number of classes, and system factors (*e.g.*, number of devices or subjects). The setup is further challenged by the relatively small number of samples available for fine-tuning (EPILEPSY: 60; FD-B: 60; GESTURE: 480; EMG: 122). Further details are in Appendix F.

**Results.** The results for the four scenarios are shown in Table 1 and Tables 4-6. Overall, our TF-C model has won 16 out of 24 tests (6 metrics in 4 scenarios) and is the second-best performer in only 8 other tests. We report all metrics but discuss the F1 score in the following. On average, our TF-C model claims a large margin of 15.4% over all baselines. Although the strongest baseline is varying (such as TS-TCC in Scenario 2; Mixing-up in Scenario 3), our model outperforms the strongest baselines by 1.5% across all scenarios. Specifically, as shown in Table 1 (HAR → GESTURE; Scenario 3), TF-C achieves the highest performance of 79.91% in F1 score, which yields a margin of 7.2% over the best baseline TS-TCC (74.57%). One potential explanation is that Scenario 3 involves a complex dataset (HAR has 6 classes while GESTURE has 8 classes) that can be difficult to model. The complexity of Scenario 3 is further verified by poor performance of all models (±80%) relative to performance on other Scenarios (±90%): TF-C shows strong robustness by

**Table 2: One-to-many pre-training evaluation.** Pre-training is performed on SLEEPEEG, followed by an independent fine-tuning on EPILEPSY, FD-B, GESTURE, and EMG.

| Scenarios | Models | Accuracy | Precision | Recall | F1 score | AUROC | AUPRC |
|---|---|---|---|---|---|---|---|
| | **Non-DL (KNN)** | $0.8525_{\pm0.0000}$ | $0.8639_{\pm0.0000}$ | $0.6431_{\pm0.0000}$ | $0.6791_{\pm0.0000}$ | $0.6434_{\pm0.0000}$ | $0.6279_{\pm0.0000}$ |
| | **Random Init.** | $0.8983_{\pm0.0656}$ | $0.9213_{\pm0.1369}$ | $0.7447_{\pm0.1135}$ | $0.7959_{\pm0.1208}$ | $0.8578_{\pm0.2153}$ | $0.6489_{\pm0.1926}$ |
| SLEEPEEG | **TS-SD** | $0.8952_{\pm0.0522}$ | $0.8018_{\pm0.2244}$ | $0.7647_{\pm0.1485}$ | $0.7767_{\pm0.1855}$ | $0.7677_{\pm0.2452}$ | $0.7940_{\pm0.1825}$ |
| ↓ | **TS2vec** | $0.9395_{\pm0.0044}$ | $0.9059_{\pm0.0116}$ | $0.9039_{\pm0.0118}$ | $0.9045_{\pm0.0067}$ | $0.9587_{\pm0.0086}$ | $0.9430_{\pm0.0103}$ |
| EPILEPSY | **CLOCS** | $\mathbf{0.9507_{\pm0.0027}}$ | $0.9301_{\pm0.0067}$ | $\mathbf{0.9127_{\pm0.0165}}$ | $\mathbf{0.9206_{\pm0.0066}}$ | $0.9803_{\pm0.0023}$ | $0.9609_{\pm0.0116}$ |
| | **Mixing-up** | $0.8021_{\pm0.0000}$ | $0.4011_{\pm0.0000}$ | $0.5000_{\pm0.0000}$ | $0.4451_{\pm0.0000}$ | $0.9743_{\pm0.0081}$ | $0.9618_{\pm0.0104}$ |
| | **TS-TCC** | $0.9253_{\pm0.0098}$ | $0.9451_{\pm0.0049}$ | $0.8181_{\pm0.0257}$ | $0.8633_{\pm0.0215}$ | $0.9842_{\pm0.0034}$ | $0.9744_{\pm0.0043}$ |
| | **SimCLR** | $0.9071_{\pm0.0344}$ | $0.9221_{\pm0.0166}$ | $0.7864_{\pm0.1071}$ | $0.8178_{\pm0.0998}$ | $0.9045_{\pm0.0539}$ | $0.9128_{\pm0.0205}$ |
| | **TF-C (Ours)** | $0.9495_{\pm0.0249}$ | $\mathbf{0.9456_{\pm0.0108}}$ | $0.8908_{\pm0.0216}$ | $0.9149_{\pm0.0534}$ | $\mathbf{0.9811_{\pm0.0237}}$ | $\mathbf{0.9703_{\pm0.0199}}$ |
| | **Non-DL (KNN)** | $0.4473_{\pm0.0000}$ | $0.2847_{\pm0.0000}$ | $0.3275_{\pm0.0000}$ | $0.2284_{\pm0.0000}$ | $0.4946_{\pm0.0000}$ | $0.3308_{\pm0.0000}$ |
| | **Random Init.** | $0.4736_{\pm0.0623}$ | $0.4829_{\pm0.0529}$ | $0.5235_{\pm0.1023}$ | $0.4911_{\pm0.0590}$ | $0.7864_{\pm0.0349}$ | $0.7528_{\pm0.0254}$ |
| SLEEPEEG | **TS-SD** | $0.5566_{\pm0.0210}$ | $0.5710_{\pm0.0535}$ | $0.6054_{\pm0.0272}$ | $0.5703_{\pm0.0328}$ | $0.7196_{\pm0.0113}$ | $0.5693_{\pm0.0532}$ |
| ↓ | **TS2vec** | $0.4790_{\pm0.0113}$ | $0.4339_{\pm0.0092}$ | $0.4842_{\pm0.0197}$ | $0.4389_{\pm0.0107}$ | $0.6463_{\pm0.0130}$ | $0.4442_{\pm0.0162}$ |
| FD-B | **CLOCS** | $0.4927_{\pm0.0310}$ | $0.4824_{\pm0.0316}$ | $0.5873_{\pm0.0387}$ | $0.4746_{\pm0.0485}$ | $0.6992_{\pm0.0099}$ | $0.5501_{\pm0.0365}$ |
| | **Mixing-up** | $0.6789_{\pm0.0246}$ | $0.7146_{\pm0.0343}$ | $\mathbf{0.7613_{\pm0.0198}}$ | $0.7273_{\pm0.0228}$ | $0.8209_{\pm0.0035}$ | $0.7707_{\pm0.0042}$ |
| | **TS-TCC** | $0.5499_{\pm0.0220}$ | $0.5279_{\pm0.0293}$ | $0.6396_{\pm0.0178}$ | $0.5418_{\pm0.0338}$ | $0.7329_{\pm0.0203}$ | $0.5824_{\pm0.0468}$ |
| | **SimCLR** | $0.4917_{\pm0.0437}$ | $0.5446_{\pm0.1024}$ | $0.4760_{\pm0.0885}$ | $0.4224_{\pm0.1138}$ | $0.6619_{\pm0.0219}$ | $0.5009_{\pm0.0477}$ |
| | **TF-C (Ours)** | $\mathbf{0.6938_{\pm0.0231}}$ | $\mathbf{0.7559_{\pm0.0349}}$ | $0.7202_{\pm0.0257}$ | $\mathbf{0.7487_{\pm0.0268}}$ | $\mathbf{0.8965_{\pm0.0135}}$ | $\mathbf{0.7871_{\pm0.0267}}$ |
| | **Non-DL (KNN)** | $0.6833_{\pm0.0000}$ | $0.6501_{\pm0.0000}$ | $0.6833_{\pm0.0000}$ | $0.6443_{\pm0.0000}$ | $0.8190_{\pm0.0000}$ | $0.5232_{\pm0.0000}$ |
| | **Random Init.** | $0.4219_{\pm0.0629}$ | $0.4751_{\pm0.0175}$ | $0.4963_{\pm0.0679}$ | $0.4886_{\pm0.0459}$ | $0.7129_{\pm0.0166}$ | $0.3358_{\pm0.1439}$ |
| SLEEPEEG | **TS-SD** | $0.6922_{\pm0.0444}$ | $0.6698_{\pm0.0472}$ | $0.6867_{\pm0.0488}$ | $0.6656_{\pm0.0443}$ | $0.8725_{\pm0.0324}$ | $0.6185_{\pm0.0966}$ |
| ↓ | **TS2vec** | $0.6917_{\pm0.0333}$ | $0.6545_{\pm0.0358}$ | $0.6854_{\pm0.0349}$ | $0.6570_{\pm0.0392}$ | $0.8968_{\pm0.0123}$ | $0.6989_{\pm0.0346}$ |
| GESTURE | **CLOCS** | $0.4433_{\pm0.0518}$ | $0.4237_{\pm0.0794}$ | $0.4433_{\pm0.0518}$ | $0.4014_{\pm0.0602}$ | $0.8073_{\pm0.0109}$ | $0.4460_{\pm0.0384}$ |
| | **Mixing-up** | $0.6933_{\pm0.0231}$ | $0.6719_{\pm0.0232}$ | $0.6933_{\pm0.0231}$ | $0.6497_{\pm0.0306}$ | $0.8915_{\pm0.0261}$ | $0.7279_{\pm0.0558}$ |
| | **TS-TCC** | $0.7188_{\pm0.0349}$ | $0.7135_{\pm0.0352}$ | $0.7167_{\pm0.0373}$ | $0.6984_{\pm0.0360}$ | $0.9099_{\pm0.0085}$ | $0.7675_{\pm0.0201}$ |
| | **SimCLR** | $0.4804_{\pm0.0594}$ | $0.5946_{\pm0.1623}$ | $0.5411_{\pm0.1946}$ | $0.4955_{\pm0.1870}$ | $0.8131_{\pm0.0521}$ | $0.5076_{\pm0.1588}$ |
| | **TF-C (Ours)** | $\mathbf{0.7642_{\pm0.0196}}$ | $\mathbf{0.7731_{\pm0.0355}}$ | $\mathbf{0.7429_{\pm0.0268}}$ | $\mathbf{0.7572_{\pm0.0311}}$ | $\mathbf{0.9238_{\pm0.0159}}$ | $\mathbf{0.7961_{\pm0.0109}}$ |
| | **Non-DL (KNN)** | $0.4390_{\pm0.0000}$ | $0.3772_{\pm0.0000}$ | $0.5143_{\pm0.0000}$ | $0.3979_{\pm0.0000}$ | $0.6025_{\pm0.0000}$ | $0.4084_{\pm0.0000}$ |
| | **Random Init.** | $0.7780_{\pm0.0729}$ | $0.5909_{\pm0.0625}$ | $0.6667_{\pm0.0135}$ | $0.6238_{\pm0.0267}$ | $0.9109_{\pm0.1239}$ | $0.7771_{\pm0.1427}$ |
| SLEEPEEG | **TS-SD** | $0.4606_{\pm0.0000}$ | $0.1545_{\pm0.0000}$ | $0.3333_{\pm0.0000}$ | $0.2111_{\pm0.0000}$ | $0.5005_{\pm0.0126}$ | $0.3775_{\pm0.0110}$ |
| ↓ | **TS2vec** | $0.7854_{\pm0.0318}$ | $\mathbf{0.8040_{\pm0.0750}}$ | $0.6785_{\pm0.0396}$ | $0.6766_{\pm0.0501}$ | $\mathbf{0.9331_{\pm0.0164}}$ | $\mathbf{0.8436_{\pm0.0372}}$ |
| EMG | **CLOCS** | $0.6985_{\pm0.0323}$ | $0.5306_{\pm0.0750}$ | $0.5354_{\pm0.0291}$ | $0.5139_{\pm0.0409}$ | $0.7923_{\pm0.0573}$ | $0.6484_{\pm0.0680}$ |
| | **Mixing-up** | $0.3024_{\pm0.0534}$ | $0.1099_{\pm0.0126}$ | $0.2583_{\pm0.0456}$ | $0.1541_{\pm0.0204}$ | $0.4506_{\pm0.1718}$ | $0.3660_{\pm0.1635}$ |
| | **TS-TCC** | $0.7889_{\pm0.0192}$ | $0.5851_{\pm0.0974}$ | $0.6310_{\pm0.0991}$ | $0.5904_{\pm0.0952}$ | $0.8851_{\pm0.0113}$ | $0.7939_{\pm0.0386}$ |
| | **SimCLR** | $0.6146_{\pm0.0582}$ | $0.5361_{\pm0.1724}$ | $0.4990_{\pm0.1214}$ | $0.4708_{\pm0.1486}$ | $0.7799_{\pm0.1344}$ | $0.6392_{\pm0.1596}$ |
| | **TF-C (Ours)** | $\mathbf{0.8171_{\pm0.0287}}$ | $0.7265_{\pm0.0353}$ | $\mathbf{0.8159_{\pm0.0289}}$ | $\mathbf{0.7683_{\pm0.0311}}$ | $0.9152_{\pm0.0211}$ | $0.8329_{\pm0.0137}$ |

learning more generalizable representations. Additionally, we visualize the learned representations in time-frequency space (Appendix I), and the analyses provide further support for the TF-C property.

## 5.2 Results: One-to-Many Pre-Training Evaluation

**Setup.** In one-to-many evaluation, pre-training is done using *one* dataset followed by fine-tuning on *multiple* target datasets independently without starting pre-training from scratch. Out of eight datasets, SLEEPEEG has most complex temporal dynamics [69] and is the largest (371,055 samples). For that reason, we pre-train a model on SLEEPEEG and separately fine-tune a well-pre-trained model on EPILEPSY, FD-B, GESTURE, and EMG.

**Results.** Results are shown in Table 2. As there are fewer commonalities between EEG signals vs. vibration, and acceleration vs. EMG, we expect that transfer learning will be less effective for them than one-to-one evaluations. The pre-training and fine-tuning datasets are largely different in the bottom three blocks (SLEEPEEG → {FD-B, GESTURE, EMG}). The large gap reasonably leads to a deterioration in baseline performances, however, our model has a noticeably higher tolerance to knowledge transfer across datasets with large gaps. Notably, We find that the proposed model with TF-C earned the best performance in 14 out of 18 settings in the three challenging settings: indicating our TF-C assumption is universal in time series. For example, our approach outperforms the strongest baseline by 8.4% (in precision) when fine-tuning on GESTURE. Our model has great potential to serve as a universal model when there is no large pre-training dataset that is similar to the small fine-tuning dataset. Furthermore, the TF-C consistently outperforms KNN and Random Init. (which are not pre-trained) by a large margin of 42.8% and 25.1% (both in F1 score) on average.

**Ablation study.** We evaluate how relevant the model components are for effective pre-training. As shown in Table 9 (SLEEPEEG → GESTURE; Appendix H), removing $\mathcal{L}_C$, $\mathcal{L}_T$, and $\mathcal{L}_F$ result in performance degradation (precision) of 6.1%, 7.2%, and 6.7%, respectively. To validate that

the performance increment is not solely brought by a third loss term no matter what consistency it measures, we replaced consistency loss $\mathcal{L}_\mathrm{C}$ with a loss term measuring the consistency within time space (named $\mathcal{L}_\mathrm{TT\text{-}C}$) or within frequency space (named $\mathcal{L}_\mathrm{FF\text{-}C}$). Results show our consistency loss outperforms $\mathcal{L}_\mathrm{TT\text{-}C}$ and $\mathcal{L}_\mathrm{FF\text{-}C}$ by 5.3% and 7.2% (accuracy), respectively.

### 5.3 Additional Downstream Tasks: Clustering and Anomaly Detection

**Clustering Task.** We evaluate the clustering performance of TF-C taking SLEEPEEG → EPILEPSY as an example. Specifically, we added a K-means (K=2), as Epilepsy has 2 classes, on top of $z_i^\mathrm{tune}$ in fine-tuning. We adopt commonly used evaluation metrics: Silhouette score, Adjusted Rand Index (ARI), and Normalized Mutual Information (NMI). Table 7 shows our TF-C obtains the best clustering surpassing the strongest baseline (TS-TCC) by a large margin (5.4% in Silhouette score). It conveys that TF-C can capture more distinctive representations with the knowledge transferred from pre-training, which is consistent with the superiority of TF-C in the above classification tasks.

**Anomaly Detection Task.** We assess how TF-C performs on a sample-level anomaly detection task. Note we work on the sample-level rather than the observation-level anomaly detection. Based on global patterns, the former aims to detect abnormal time series samples instead of outlier observations in a sample (as in BTSF [50] and USAD [70]) which emphasizes local context. Specifically, In the scenario of FD-A → FD-B, we built a small subset of FD-B with 1,000 samples, of which 900 are from undamaged bearings, and the remaining 100 are from bearings with inner or outer damage. Undamaged samples are considered "normal," and inner/outer damaged samples are "outliers." In fine-tuning, we used one-class SVM on top of learned representations $z_i^\mathrm{tune}$. The experimental results (Table 8) show that our TF-C outperforms five competitive baselines with 4.5% in F-1 Score. Results show that the proposed TF-C is more sensitive to anomalous samples and can effectively detect the abnormal status in mechanical devices.

## 6 Conclusion

We develop a pre-training approach that introduces time-frequency consistency (TF-C) as a mechanism to support knowledge transfer between time-series datasets. The approach uses self-supervised contrastive estimation and injects TF-C into pre-training, bringing time-based and frequency-based representations and their local neighborhoods close together in the latent space.

**Limitations and future directions.** TF-C property can serve as a universal property for pre-training on diverse time series datasets. Additional generalizable properties, such as temporal autoregressive processes, could also be helpful for pre-training on time series. Further, while our method expects as input a regularly sampled time series, it can handle irregularly sampled time series by using an encoder (such as Raindrop [71] and SeFT [72]) that can embed irregular time series. For frequency encoder inputs $x_i^\mathrm{F}$, alternatives include resampling or interpolation to obtain regularly sampled signals and using regular or non-uniform FFT operations. Furthermore, TF-C's current embedding strategy and loss functions are favorable for classification, leveraging global information over tasks that use local context (*e.g.*, forecasting). Results show that the TF-C approach performs well across broad downstream tasks, including classification, clustering, and anomaly detection (Sec. 5.3).

## Acknowledgments and Disclosure of Funding

We gratefully acknowledge support by US Air Force Contract No. FA8702-15-D-0001, Harvard Data Science Initiative, and awards from Amazon Research, Bayer Early Excellence in Science, AstraZeneca Research, and Roche Alliance with Distinguished Scientists. T.T. is supported by the Under Secretary of Defense for Research and Engineering under US Air Force Contract No. FA8702-15-D-0001. Any opinions, findings, conclusions or recommendations expressed in this material are those of the authors and do not necessarily reflect the views of the funders.

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

## Broader Impacts

Our approach for self-supervised pre-training improves classification performance on target datasets in different application scenarios. The recognition of time-frequency consistency as a universal property specific to time series data is a weak assumption that enables effective, task- and domain-agnostic transfer learning. We believe our work will inspire the research community to uncover other universal properties for transfer learning. We also hope our work will also attract more researchers to the more general problem of time series representation learning which is still underappreciated relative to problems from CV and NLP fields.

On the society level, our work, along the line of transfer learning, can facilitate more efficient use of time series data in various settings. For example, in medical settings, some diseases of clinical interest may have very small labelled dataset. In this case, unlabelled data from patients of different diseases but with similar underlying physiological conditions can be used to pre-train the model. However, practitioners need to be aware of the limitations of the model, including that it may make biased predictions. Specifically, bias may exist in the source dataset used for pre-training due to an imbalance of samples from subjects of different demographic attributes. Also, the standardized medical protocols for collecting these datasets might be unsuitable for subjects with certain physiological attributes, creating unforeseen bias that may be transferred to fine-tuning.

All datasets in this paper are publicly available and are not associated with any privacy or security concern. Furthermore, we have followed guidelines on responsible use specified by primary authors of the datasets used in the current work.

