## Appendix A  Further information on the relationship between our pre-training approach and domain adaptation

Here we note our problem definition of pre-training is fundamentally different from domain adaptation [S1, S2, S3, S4, S5, S6][1] in order to prevent any confusion between this work and domain adaptation methods. DA applies a model trained on a pre-training dataset (*i.e.*, source dataset) to a different target dataset [21, 42]. In contrast, self-supervised pre-training has four key differences with domain adaptation. (1) First, our model only requires the pre-training dataset while domain adaptation techniques generally require access to the target dataset [S7, S8, S9, S10, S11, S12, S13, S14, S15, S16]. (2) Second, our model can be applied to multiple unseen target datasets (without re-training the pre-trained model for every target dataset) while domain adaptation approaches use the target dataset during model training, *e.g.*, [S17, S18, S19, S20, S12, S14, S21] (see also Sec. 5.2 for experimental results). (3) Third, our approach can be used in scenarios where the feature space in pre-training is different from that in the target dataset (see Scenarios 1, 3, and 4 in Sec. 5.1 for experimental results). In contrast, domain adaptation methods usually restrict pre-training and target datasets to have the same feature space (but possible different distributions), *e.g.*, [S22, S18, S19, S20, S13].

In summary, to support transfer learning across different time series datasets, a pre-training approach needs a capability to capture a generalizable property of time series, one that is shared across different time series datasets regardless of the specific semantic meaning of a time series signal (*e.g.*, ECG, EMG, acceleration, vibration), conditions of data acquisition (*e.g.*, variation across subjects and devices), sampling frequencies, etc. This work develops a self-supervised contrastive pre-training strategy that fulfills these requirements by injecting an appropriate inductive bias (called Time-Frequency Consistency, TF-C, into the model (Sec. 3).

Further, we clarify that the term 'self-supervised' has different meanings in DA and in pre-training [S23, S24, S25, S26]. The 'self-supervised domain adaptation' [S27, S16, S21, S15] or 'unsupervised domain adaptation' [S1, S22, S28, S11, S14] means that there are no labels in the target dataset, however that still requires labels in the pre-training dataset. In contrast, 'self-supervised pre-training' [S29, S30, S31] (*i.e.*, the problem studied here, in line with a breadth of existing literature on pre-training) indicates the setting where no labels are available in pre-training.

## Appendix B  Detailed differences with CoST and BTSF

Up to the submission of this manuscript, there is no existing contrastive augmentations in time series' frequency domain. There are two models, CoST [49] and BTSF [50], that involved frequency domain in contrastive learning, however, the proposed TF-C is fundamentally different with them in the following aspects. We take BTSF as an example while the differences also apply to CoST.

- Problem definitions for both papers are different. Our method is designed to produce generalizable representations that can transfer to a different time series dataset (going from pre-training to a fine-tuning dataset) for the purpose of transfer learning. In contrast, BTSF attempts to learn embeddings within the same dataset for the purpose of representation learning. Our model captures the TF-C property invariant to different time series (in terms of various temporal dynamics, semantic meaning, etc.) and can thus serve as a vehicle for transfer learning. In contrast, BTSF learns embeddings invariant to perturbations (i.e., instance-level dropout) of the same time series.

- The modeling of the frequency domain is different in both papers. We developed augmentations in the frequency domain based on the spectral properties of time series. In contrast, although BTSF involves a frequency domain, its data transformation is solely implemented in the time domain (using instance-level dropout; Sec. 3.1 in BTSF). That is, the BTSF method applies the FFT after augmenting samples in the time domain which can lead to information loss.

- BTSF emphasizes fusing temporal and spectral features to generate discriminative embeddings. Unlike BTSF, our model leverages the consistency between time-based and frequency-based embeddings to produce generalizable time series representations. Our

---

[1]The supplementary document contains additional references, prefixed by 'S'. These additional references are listed at the end of the Appendix.

model maps every sample to a time-frequency embedding space and constrains the relative relationships between embeddings (through triplet loss) according to the TF-C property. The underlying consistency allows TF-C to realize transfer learning across time series datasets, which is TF-C's unique advantage.

In summary, TF-C introduces frequency domain augmentations in the sense that it directly perturbs the frequency spectrum. TF-C is the first method that uses frequency domain augmentations to enable transfer learning in time series.

## Appendix C  Time series invariance between time and frequency domains

Here, we provide an analogy, from images to time series, to aid understanding of the 'invariance' property of data representations.

In computer vision, it is well known that an image and its augmented views obtained by simple transformations (e.g., rotation, translation, scaling, etc.) can be used in different frameworks such as transformation prediction and contrastive instance discrimination to obtain invariant representations. These transformations are used as augmentations to guide the learning of self-supervised representations. For transformation prediction, the model learns representations that are equivariant to the selected transformation as the information embedded in the transformation needs to be embedded in the representation for the final layer to solve the pretext task. For contrastive instance discrimination, the representations are sensitive to the instances while learning invariances to transformations or views. The intuition behind such transformations is that the underlying object in the image is the same no matter how the image is rotated, translated, re-scaled, etc. In other words, the information carried by the original image and the transformed image is the same.

Similarly, the time domain and frequency domain representations carry the same information in time series. Thus it is reasonable to expect that by exploring local neighborhoods in the time domain and frequency domain and enforcing consistency of feature representations inter- and intra- domains, invariance properties are captured in our model via self-supervised learning. Thus, information carried in time and frequency domains of the same or similar time series sample should be the same. This invariance, formalized as 'Time-Frequency Consistency', is helpful for self-supervised pre-training. We will include the above discussion in the camera-ready version.

## Appendix D  Additional information on datasets and pre-training evaluation

### D.1  Datasets

We use eight diverse time series datasets to evaluate our model. The datasets used in one-to-one and one-to-many pre-training evaluations are the same. The dataset statistics are shown in Table 3. Processed model-ready datasets are in our GitHub Repository (https://anonymous.4open.science/r/TFC-pre-training-6B07). Following is a detailed description of datasets.

SLEEPEEG [61]. The dataset contains 153 whole-night sleeping electroencephalography (EEG) recordings produced by a sleep cassette. Data are collected from 82 healthy subjects. The 1-lead EEG signal is sampled at 100 Hz. We segment the EEG signals into segments (window size is 200) without overlapping, and each segment forms a sample. Every sample is associated with one of the five sleeping patterns/stages: Wake (W), Non-rapid eye movement (N1, N2, N3), and Rapid Eye Movement (REM). After segmentation, we have 371,055 EEG samples. The raw dataset (https://www.physionet.org/content/sleep-edfx/1.0.0/) is distributed under the Open Data Commons Attribution License v1.0.

EPILEPSY [62]. The dataset contains single-channel EEG measurements from 500 subjects. For every subject, the brain activity was recorded for 23.6 seconds. The dataset was then divided and shuffled (to mitigate sample-subject association) into 11,500 samples of 1 second each, sampled at 178 Hz. The raw dataset features five classification labels corresponding to different states of subjects or measurement locations — eyes open, eyes closed, EEG measured in the healthy brain region, EEG measured in the tumor region, and whether the subject has a seizure episode. To emphasize the distinction between positive and negative samples in terms of epilepsy, We merge the first four classes into one, and each time series sample has a binary label describing if the

**Table 3:** Description of datasets use in the four different application scenarios. We have two datasets (a pre-training dataset and a fine-tuning dataset) in each scenario. For the number of samples in fine-tuning dataset, "A/B/C" denotes we use A samples for fine-tuning, B samples for validation, and C samples for test. To test our effectiveness on small datasets (which is practically meaningful), we limit the fine-tuning set to a very small set with less than 320 samples. We ensure the fine-tuning set is balanced in terms of classes. See D.2 for more details on varying lengths of datasets.

| Scenario # | | Dataset | # Samples | # Channels | # Classes | Length | Freq (Hz) |
|---|---|---|---|---|---|---|---|
| 1 | Pre-training | SLEEPEEG | 371,055 | 1 | 5 | 200 | 100 |
| | Fine-tuning | EPILEPSY | 60/20/11,420 | 1 | 2 | 178 | 174 |
| 2 | Pre-training | FD-A | 8,184 | 1 | 3 | 5,120 | 64K |
| | Fine-tuning | FD-B | 60/21/13,559 | 1 | 3 | 5,120 | 64K |
| 3 | Pre-training | HAR | 10,299 | 9 | 6 | 128 | 50 |
| | Fine-tuning | GESTURE | 320/120/120 | 3 | 8 | 315 | 100 |
| 4 | Pre-training | ECG | 43,673 | 1 | 4 | 1,500 | 300 |
| | Fine-tuning | EMG | 122/41/41 | 1 | 3 | 1,500 | 4,000 |

associated subject is experiencing a seizure or not. There are 11,500 EEG samples in total. To evaluate the performance of a pre-trained model on a small fine-tuning dataset, we choose a small set (60 samples; 30 samples for each class) for fine-tuning and assess the model with a validation set (20 samples; 10 samples for each class). Finally, the model with the best validation performance is used to make predictions on the test set (*i.e.*, the remaining 11,420 samples). Statistics of fine-tuning, validation, and test sets in the other three target datasets are in Appendix 3. The raw dataset (https://repositori.upf.edu/handle/10230/42894) is distributed under the Creative Commons License (CC-BY) 4.0.

**FD-A** and **FD-B** [63]. The dataset is generated by an electromechanical drive system that monitors the condition of rolling bearings and detects their failures. Four subsets of data are collected under various conditions, whose parameters include rotational speed, load torque, and radial force. Each rolling bearing can be undamaged, inner damaged, and outer damaged, which leads to three classes in total. We denote the subsets corresponding to condition A and condition B as Faulty Detection Condition A (**FD-A**) and Faulty Detection Condition B (**FD-B**), respectively. Each original recording has a single channel with a sampling frequency of 64k Hz and lasts 4 seconds. To deal with lengthy recordings, we follow the procedure described by Eldele *et al.* [48]. Specifically, we use a sliding window length of 5,120 observations and a shifting length of 1,024 or 4,096 to ensure that samples are relatively balanced between classes. The raw dataset (https://mb.uni-paderborn.de/en/kat/main-research/datacenter/bearing-datacenter/data-sets-and-download) is distributed under the Creative Commons Attribution-Non Commercial 4.0 International License.

**HAR** [64]. This dataset contains recordings of 30 health volunteers performing daily activities, including walking, walking upstairs, walking downstairs, sitting, standing, and lying. Prediction labels are the six activities. The wearable sensors on a smartphone measure triaxial linear acceleration and triaxial angular velocity at 50 Hz. After preprocessing and isolating gravitational acceleration from body acceleration, there are nine channels (*i.e.*, 3-axis accelerometer, 3-axis gyroscope, and 3-axis magnetometer) in total. The raw dataset (https://archive.ics.uci.edu/ml/datasets/Human+Activity+Recognition+Using+Smartphones) is distributed as-is. Any commercial use is not allowed.

**GESTURE** [65]. The dataset contains accelerometer measurements of eight simple gestures that differ based on the paths of hand movement. The eight gestures are: hand swiping left, right, up, and down, hand waving in a counterclockwise or clockwise circle, hand waving in a square, and waving a right arrow. The classification labels are these eight different kinds of gestures. The original paper reports the inclusion of 4,480 gesture samples, but through the UCR database, we can only recover 440 samples. The dataset is balanced, with 55 samples in each class, and is of an appropriate size for the fine-tuning. The original paper does not explicitly report sampling frequency but is presumably 100 Hz. The dataset uses three channels corresponding to three coordinate directions of acceleration. Note, when transferring knowledge from **HAR** (nine channels) to **GESTURE** (three channels): we use all the nine channels in pre-training if the model (*i.e.*, TF-C, CLOCS, SimCLR, and TS-TCC) has channel generalization ability (*i.e.*, allows different channel numbers in pre-training and fine-tuning datasets); otherwise, if the model (*i.e.*, KNN, TS-SD, Mixing-up, and TS2vec) don't have channel generalization

**Table 4:** Performance on one-to-one setting (scenario 1): pre-training on **SLEEPEEG** and fine-tuning on **EPILEPSY**.

| Models | Accuracy | Precision | Recall | F1 score | AUROC | AUPRC |
|---|---|---|---|---|---|---|
| **Non-DL (KNN)** | $0.8525_{\pm0.0000}$ | $0.8639_{\pm0.0000}$ | $0.6431_{\pm0.0000}$ | $0.6791_{\pm0.0000}$ | $0.6434_{\pm0.0000}$ | $0.6279_{\pm0.0000}$ |
| **Random Init.** | $0.8983_{\pm0.0656}$ | $0.9213_{\pm0.1369}$ | $0.7447_{\pm0.1135}$ | $0.7959_{\pm0.1208}$ | $0.8578_{\pm0.2153}$ | $0.6489_{\pm0.1926}$ |
| **TS-SD** | $0.8952_{\pm0.0522}$ | $0.8018_{\pm0.2244}$ | $0.7647_{\pm0.1485}$ | $0.7767_{\pm0.1855}$ | $0.7677_{\pm0.2452}$ | $0.7940_{\pm0.1825}$ |
| **TS2vec** | $0.9395_{\pm0.0044}$ | $0.9059_{\pm0.0116}$ | $0.9039_{\pm0.0118}$ | $0.9045_{\pm0.0067}$ | $0.9587_{\pm0.0086}$ | $0.9430_{\pm0.0103}$ |
| **CLOCS** | $\mathbf{0.9507}_{\pm\mathbf{0.0027}}$ | $0.9301_{\pm0.0067}$ | $\mathbf{0.9127}_{\pm\mathbf{0.0165}}$ | $\mathbf{0.9206}_{\pm\mathbf{0.0066}}$ | $0.9803_{\pm0.0023}$ | $0.9609_{\pm0.0116}$ |
| **Mixing-up** | $0.8021_{\pm0.0000}$ | $0.4011_{\pm0.0000}$ | $0.5000_{\pm0.0000}$ | $0.4451_{\pm0.0000}$ | $0.9743_{\pm0.0081}$ | $0.9618_{\pm0.0104}$ |
| **TS-TCC** | $0.9253_{\pm0.0098}$ | $0.9451_{\pm0.0049}$ | $0.8181_{\pm0.0257}$ | $0.8633_{\pm0.0215}$ | $0.9842_{\pm0.0034}$ | $0.9744_{\pm0.0043}$ |
| **SimCLR** | $0.9071_{\pm0.0344}$ | $0.9221_{\pm0.0166}$ | $0.7864_{\pm0.1071}$ | $0.8178_{\pm0.0998}$ | $0.9045_{\pm0.0539}$ | $0.9128_{\pm0.0205}$ |
| **TNC** | $0.6701_{\pm0.0071}$ | $0.5336_{\pm0.2742}$ | $0.5011_{\pm0.0016}$ | $0.4024_{\pm0.0033}$ | $0.5853_{\pm0.0487}$ | $0.6276_{\pm0.0438}$ |
| **CPC** | $0.662_{\pm0.0110}$ | $0.4340_{\pm0.2215}$ | $0.5028_{\pm0.0064}$ | $0.406_{\pm0.0134}$ | $0.5722_{\pm0.0748}$ | $0.6145_{\pm0.0575}$ |
| **TF-C (Ours)** | $0.9495_{\pm0.0249}$ | $\mathbf{0.9456}_{\pm\mathbf{0.0108}}$ | $0.8908_{\pm0.0216}$ | $0.9149_{\pm0.0534}$ | $\mathbf{0.9811}_{\pm\mathbf{0.0237}}$ | $\mathbf{0.9703}_{\pm\mathbf{0.0199}}$ |

ability, we only use three acceleration channels in **HAR** for pre-training. The raw dataset is accessible through http://www.timeseriesclassification.com/description.php?Dataset=UWaveGestureLibrary. While the distribution license is not explicitly mentioned, the dataset is a public resource based on [65].

**ECG** [26]. This is the 2017 PhysioNet Challenge focusing on classifying ECG recordings. The single-lead ECG measures four different underlying conditions of cardiac arrhythmias. More specifically, these classes correspond to the recordings of normal sinus rhythm, atrial fibrillation (AF), alternative rhythm, or others (too noisy to be classified). The recordings are sampled at 300 Hz. Furthermore, the dataset is imbalanced, with much fewer samples from the atrial fibrillation and noisy classes out of all four. To preprocess the dataset, we use the code from the CLOCS paper, which applied a fixed-length window of 1,500 observations to divide long recordings into short samples of 5 seconds that are physiologically meaningful. Because of the imbalanced dataset, we report AUROC and AUPRC (insensitive to label distribution) in our results. The raw dataset (https://physionet.org/content/challenge-2017/1.0.0/) is distributed under the Open Data Commons Attribution License v1.0.

**EMG** [66]. Electromyograms (EMGs) measure muscle responses as electrical activity in response to neural stimulation, and they can be used to diagnose certain muscular dystrophies and neuropathies. The dataset consists of single-channel EMG recording from the tibialis anterior muscle of three healthy volunteers suffering from neuropathy and myopathy. The recordings are sampled at a frequency of 4K Hz. Each patient (*i.e.*, disorder) is a classification category. Then the recordings are split into time-series samples using a fixed-length window of 1,500 observations. The raw dataset (https://physionet.org/content/emgdb/1.0.0/) is distributed under the Open Data Commons Attribution License v1.0.

### D.2 Varying lengths and dimensionality of time series in pre-training vs. target datasets

We use established strategies to address the challenging question of how to process time series with the variable number of dimensions (channels) and measurements (note that this study focuses on designing a pre-training model that can transfer knowledge across disparate time-series datasets). When the dimensionality or length of time series is different between pre-training and target datasets, we view the pre-training dataset as an anchor according to which we adjust (pre-process) the target dataset. For length adjustments, we use zero padding to increase the number of observations or use clipping to reduce the number of observations. In the four Scenarios, we align the time series lengths to 178, 5120, 206, and 1500, respectively. For Scenario 3, as there's a large gap between 128 and 315, assertive padding or clipping could loss much information, thus we take a trade-off by padding 128 to 206 while trimming 315 to 206. The users are feel to select any other length in their experiments. For dimensionality adjustments, our model applies to time series at the level of a single lead, enabling the ability to generalize to multivariate time series when we make different channels share the model (*i.e.*, time encoder, frequency encoder, and two projectors) parameters. In this way, we have a $z_i^{\text{TUNE}}$ for each channel of the time series (Sec. 4) and concatenate the learned representations across all channels to form the final representation of the input time series sample.

**Table 5:** Performance on one-to-one setting (scenario 2): pre-training on **FD-A** and fine-tuning on **FD-B**.

| Models | Accuracy | Precision | Recall | F1 score | AUROC | AUPRC |
|---|---|---|---|---|---|---|
| **Non-DL (KNN)** | $0.4473_{\pm0.0000}$ | $0.2846_{\pm0.0000}$ | $0.3275_{\pm0.0000}$ | $0.2284_{\pm0.0000}$ | $0.4946_{\pm0.0000}$ | $0.3307_{\pm0.0000}$ |
| **Random Init.** | $0.4736_{\pm0.1085}$ | $0.4829_{\pm0.1235}$ | $0.5235_{\pm0.0956}$ | $0.4911_{\pm0.1123}$ | $0.7864_{\pm0.2659}$ | $0.7528_{\pm0.1233}$ |
| **TS-SD** | $0.5465_{\pm0.0417}$ | $0.5795_{\pm0.0608}$ | $0.6613_{\pm0.0324}$ | $0.5567_{\pm0.0478}$ | $0.7148_{\pm0.0014}$ | $0.6370_{\pm0.0231}$ |
| **TS2vec** | $0.6494_{\pm0.0379}$ | $0.6614_{\pm0.0599}$ | $0.7260_{\pm0.0377}$ | $0.6797_{\pm0.0553}$ | $0.8089_{\pm0.0254}$ | $0.7138_{\pm0.0593}$ |
| **CLOCS** | $0.7118_{\pm0.0361}$ | $0.7607_{\pm0.0830}$ | $0.7877_{\pm0.0294}$ | $0.7638_{\pm0.0696}$ | $0.8161_{\pm0.0146}$ | $0.8133_{\pm0.0295}$ |
| **Mixing-up** | $0.7821_{\pm0.0110}$ | $0.8887_{\pm0.0039}$ | $0.8404_{\pm0.0081}$ | $0.8307_{\pm0.0104}$ | $0.9339_{\pm0.0003}$ | $0.9348_{\pm0.0010}$ |
| **TS-TCC** | $0.8497_{\pm0.0114}$ | $0.8906_{\pm0.0087}$ | $\mathbf{0.8899_{\pm0.0083}}$ | $0.8898_{\pm0.0083}$ | $0.9145_{\pm0.0138}$ | $0.8957_{\pm0.0198}$ |
| **SimCLR** | $0.5439_{\pm0.0562}$ | $0.6372_{\pm0.0568}$ | $0.6128_{\pm0.1344}$ | $0.5799_{\pm0.1286}$ | $0.7383_{\pm0.0622}$ | $0.6548_{\pm0.1152}$ |
| **TF-C (Ours)** | $\mathbf{0.8934_{\pm0.0379}}$ | $\mathbf{0.9209_{\pm0.0234}}$ | $0.8537_{\pm0.0486}$ | $\mathbf{0.9162_{\pm0.0826}}$ | $\mathbf{0.9435_{\pm0.0259}}$ | $\mathbf{0.9527_{\pm0.0134}}$ |

**Table 6:** Performance on one-to-one setting (scenario 4): pre-training on **ECG** and fine-tuning on **EMG**.

| Models | Accuracy | Precision | Recall | F1 score | AUROC | AUPRC |
|---|---|---|---|---|---|---|
| **Non-DL (KNN)** | $0.4390_{\pm0.0000}$ | $0.3771_{\pm0.0000}$ | $0.5143_{\pm0.0000}$ | $0.3979_{\pm0.0000}$ | $0.6025_{\pm0.0000}$ | $0.4083_{\pm0.0000}$ |
| **Random Init.** | $0.878_{\pm0.1259}$ | $0.5909_{\pm0.1135}$ | $0.6667_{\pm0.1534}$ | $0.6238_{\pm0.2315}$ | $0.9109_{\pm0.1264}$ | $0.7771_{\pm0.1359}$ |
| **TS-SD** | $0.4606_{\pm0.0000}$ | $0.1544_{\pm0.0000}$ | $0.3333_{\pm0.0000}$ | $0.2111_{\pm0.0000}$ | $0.5031_{\pm0.0219}$ | $0.3805_{\pm0.0165}$ |
| **TS2vec** | $0.9704_{\pm0.0109}$ | $0.9666_{\pm0.0186}$ | $0.9751_{\pm0.0082}$ | $\mathbf{0.9746_{\pm0.0141}}$ | $0.9948_{\pm0.0070}$ | $\mathbf{0.9780_{\pm0.0305}}$ |
| **CLOCS** | $0.8829_{\pm0.0499}$ | $0.8492_{\pm0.0620}$ | $0.8134_{\pm0.1262}$ | $0.8037_{\pm0.1080}$ | $0.9385_{\pm0.0369}$ | $0.8178_{\pm0.0945}$ |
| **Mixing-up** | $0.9121_{\pm0.0872}$ | $0.8007_{\pm0.2158}$ | $0.8518_{\pm0.1776}$ | $0.8215_{\pm0.2010}$ | $\mathbf{0.9999_{\pm0.0000}}$ | $0.9999_{\pm0.0000}$ |
| **TS-TCC** | $0.9590_{\pm0.0135}$ | $\mathbf{0.9684_{\pm0.0140}}$ | $0.8994_{\pm0.0304}$ | $0.9244_{\pm0.0247}$ | $0.9800_{\pm0.0192}$ | $0.9663_{\pm0.0348}$ |
| **SimCLR** | $0.8878_{\pm0.0218}$ | $0.8209_{\pm0.0307}$ | $0.8533_{\pm0.0433}$ | $0.8225_{\pm0.0339}$ | $0.9565_{\pm0.0153}$ | $0.8337_{\pm0.0163}$ |
| **TF-C (Ours)** | $\mathbf{0.9756_{\pm0.0071}}$ | $0.9444_{\pm0.0029}$ | $\mathbf{0.9803_{\pm0.0000}}$ | $0.9596_{\pm0.0003}$ | $0.9801_{\pm0.0012}$ | $0.8867_{\pm0.0122}$ |

## Appendix E    TF-C and baseline architectures and implementation details

We compare our TF-C model against eight state-of-the-art baselines and two additional methods (Sec. 5): (1) directly fit a KNN (K=2) classifier with the target (fine-tuning) dataset; (2) randomly initialize a model (the model structure and experimental settings are the same with our TF-C model), ignore pre-training, and directly train from scratch with the target dataset.

We implement the baselines follow the corresponding papers including TS-SD [12], TS2vec [47], CLOCS [41], Mixing-up [18], TS-TCC [48], SimCLR [40], TNC [46], and CPC [30]. We use default settings for hyper-parameters as reported in the original works unless noted below. All pre-training and fine-tuning with baselines are done with a single Tesla V100 GPU with 32 Gb of allocated memory provided by Harvard Medical School's O2 High Performance Computing platform.

**TF-C (our model)** Our time-based contrastive encoder $F_T$ adopts 1-D ResNet backbone similar to SimCLR. Specifically, after hyper-parameter tuning, the encoder consists of three layers convolutional blocks: the kernel sizes in all layers are 8; the strides are 8, 1, and 1, respectively; the the depths are 32, 64, and 128, respectively. We use max pooling after each convolutional layer and all the pooling kernel sizes and strides are set as 2. The cross-space projector $R_T$ contains two fully-connected layers with hidden dimensions 256 and 128, respectively. In the transformation from time space to frequency space, we use the full spectrum (symmetrical) thus $x^T$ and $x^F$ have the same dimension. For simplify, the frequency encoder $F_F$ and projector $R_F$ have same structure (but different parameters) with their counterparts in time space. Our preliminary experiments show that the change of model structure has relatively small change on performance. For example, replacing $F_T$ by 2-layer LSTM or 3-layer Transformer only cause a slight drop of F1 score ($\pm$ 1.2%; one-to-one setting; Scenario 1). In pre-training, we use Adam optimizer with learning rate of 0.0003 and 2-norm penalty coefficient of 0.0005. We use batch size of 64 and training epoch of 40. We record the best performance in terms of F1 score and save the parameters in the corresponding epoch as $\Theta$. In fine-tuning, we initialize the model parameters with $\Theta$ and optimize the model on fine-tuning set. The hyper-parameters are the same as in pre-training. For the classification task in fine-tuning stage, we adopt a 2-layer fully-connected layer as a classifier. The hidden dimensions are 64 and the number classes in target dataset, respectively. We measure the classification loss via the cross entropy function and the loss is jointly optimized with the fine-tuning loss (*i.e.*, $\mathcal{L}_{\text{TF-C}}$).

**TS-SD** [12] uses a modified attention mechanism to encode latent features and use denoising or DTW-similarity prediction as pretext-tasks.The encoder network is based on self-attention mechanism [68] but instead of the linear layer that produce the K, Q, V matrices from the input sequence, TS-SD uses a single convolutional layer to replace the linear layer. With multihead attention, TS-SD sets a different kernel size for each head to attempt capture short- and long-range temporal patterns. Although we found that 12 heads was relatively redundant and may be replaced by fewer heads, we still use 12 heads to keep the same with the original work in [12]. For self-supervised pre-training, we adopt the denoising pretext task that attempts to minimize the mean square distance between the original input time series and the output of the encoder, for which the input is an augmented time series sample with noise added to a subseries. To cope with small numerical values, we use a learning rate of 3e-7 in pre-training and 3e-4 during fine-tuning. While training on pre-training dataset for the four scenarios, the batch sizes varied from 4 to 128 and were manually chosen to achieve reasonable performance for each scenario. During fine-tuning, we use # epochs ranging from 20 to 80 and a batch size of 16. Although the source code of TS-SD is not released, we implement the TS-SD model to our best understanding and tune the model to achieve the best performance as a strong baseline. We public the implemented TS-SD in our online repository.

**TS2vec** [47] introduces the notion of contextual consistency and uses a hierarchical loss function to capture long-range structure in time series. TS2vec is a powerful representative learning method and have a specially-designed architecture. The encoder network consists of three components. First, the input time series is augmented by selecting overlapping subseries. They are projected into a higher dimensional latent space. Then, latent vectors for input time series are masked at randomly chosen positions. Finally, a dilated CNN with residual blocks produce the contextual representations. To compute the loss, the representations are gradually pooled along the time dimension and at each step a loss function based on contextual consistency is applied. For the baseline experiment, we found that the original 10 layers of ResNet blocks is redundant, and we reduce residual blocks in the encoder from 10-layer to 2-layer without compromising model performances. To make the model comparable with other baselines, we also restrict the hidden dimension to 1 and use a batch size of 64, except in the second scenario we reduce it to 16 given the length of samples in the **FD-A/FD-B** datasets.

**CLOCS** [41] assumes that samples from different subjects are negative pairs in contrastive learning and applies it to classify ECG signals. CLOCS makes the basic assumption that time series samples from different patients (the application scenario is in learning cardiac signals) should have dissimilar representations. Based on that, they devised three slightly different architectures - CMSC, CMLC, and CMSMLC, where ECG recordings from non-overlapping temporal segments, different ECG leads, and both of these are treated as positive pairs in case if they come from the same patients. For this model, the encoder consists of alternating convolutional and pooling layers. So given the different input lengths, the hyper-parameters for these layers have to be modified accordingly. For example, we use kernel sizes of $7, 7, 7$, stride of $3, 3, 3$, pooling layer size of $2$ for the last scenario of **ECG** to **EMG** but increased stride to $4$ and increased representation size for a time series sample from $64$ to $128$ in case of **FD-A** to **FD-B** due to much longer time series samples.

**Mixing-up** [18] proposes new mixing-up augmentation and pretext tasks that aim to correctly predict the mixing proportion of two time series samples. In Mixing-up, the augmentation is chosen as the convex combination of two randomly drawn time series from the dataset, where the mixing parameter is random drawn from a beta distribution. The contrastive loss is then computed between the two inputs and the augmented time series. The loss is a minor modification of NT-Xent loss and is designed to encourage the correct prediction of the amount of mixing. We use the same beta distribution as reported in the original Mixing-up model.

**TS-TCC** [48] leverages contextual information with a transformer-based autoregressive model and ensures transferability by using both strong and weak augmentations. TS-TCC proposed a challenging pretext task. An input time series sample is first augmented by adding noise, scaling, and permuting time series. The views are then passed through an encoder consisting of three convolutional layers before processed by the temporal contrasting module. During temporal contrasting, for each view, a transformer architecture is used to learn a contextual representation. The learned representation is then used to predict latent observation of the other augmented view at a future time. The contextual representations are then projected and maximized similarity using NT-Xent loss. For this baseline, we mostly adopted the hyper-parameters presented in the original paper. We use a learning rate of 3e-4 throughout, and a batch size of 128 during pre-training and 16 during fine-tuning.

**SimCLR** [40] is a state-of-the-art model in self-supervised representation learning of images. It utilizes deep learning architectures to generate augmentation-based embeddings and optimize the model parameters by minimizing NT-Xent loss in the embedding space. [40] applies the original SimCLR model for time series data. In this work, we compare with the modified SimCLR as in [40]. The SimCLR contrastive learning framework consists of four major components. Although initially proposed for image data, it is readily adapted to time series, as shown in [40]. An input time series sample is first stochastically augmented into two related views. Then a base encoder extracts representation vectors. ResNet is used as encoder backbone for simplicity. Then the projection head transforms representations into a latent space where the NT-Xent loss is applied. For time series, SimCLR investigated different augmentations, including adding noise, scaling, rotation, negation, flipping in time, permutation of subseries, time warping, and channel shuffling. All the unmentioned hyper-parameters are kept the same with the original model.

**TNC** [46] focuses on learning representations that encode the underlying state of a non-stationary time series. This is achieved by ensuring that the distribution of observations in the latent space is different from the distribution of temporally separated observations. For each time point, the TNC model calculates a neighborhood size using the Augmented Dickey–Fuller test. The encoded representations of two different windows will be passed to a discriminator to predict the probability that they belong to the same temporal neighborhood. We implement the TFC model following the public GitHub repository provided by the authors. We observed that the ADF tests are extremely slow, so TNC is only used as a baseline in scenario 1 in one-to-one setting (classification task).

**CPC** [30] aims to learn the representations that encode the global information shared across different segments of a time series signal by the strategy of predictive coding. Instead of modelling the conditional distribution at future time points for prediction, the authors choose to encode the context and observation into a vector representation in a way that maximally preserves their mutual information. For the model architecture, a nonlinear encoder first maps the sequence of observations to latent representations. Then an autoregressive model summarizes past latent representations to produce a context representation at every time point. The two components are trained to jointly optimize a loss function which is inspired by noise contrastive estimation. We used the implementation provided by the authors of the TNC paper [46] and kept their default choices of parameters whenever reasonable.

The TS-SD is the only method, to our knowledge, that explicitly aims at time series pre-training but the underlying assumption that can support knowledge transfer across different time series datasets is not clearly provided. The TS2vec, CLOCS, Mixing-up, TS-TCC, TNC, and CPC are designed for representation learning (instead of pre-training) but involved transfer learning setting in their experiments. We modify them to perform pre-training task and make them comparable to our model. The SimCLR is a widely use baseline in computer vision and simply adopted to deal with time series by [40]. We compare our model with SimCLR to further show our superiority.

## Appendix F    Additional results on one-to-one evaluation

In one-to-one pre-training evaluation, we consider four different scenarios with paired pre-training and fine-tuning datasets from different fields (Sec. 5.1). In each scenario, the pre-training and fine-tuning datasets share same semantic meanings (*e.g.*, both EEG signals; Scenario 1) or similar semantic meanings (*e.g.*, ECG and EMG are both physiological signals; Scenario 4). For example, in the first scenario, **SLEEPEEG** and **EPILEPSY** both involve EEG measurements but the recordings are taken under different physical conditions.

For each scenario, we first train each baseline model using the pre-training dataset and record the converged parameters. The parameters are then taken to initialize the model on the fine-tuning phase which uses the target dataset. In fine-tuning, we allowed all parameters in the model to be optimized (no freezing of earlier layers; full fine-tuning). Our preliminary results show that the performance of full fine-tuning is slightly better than partial fine-tuning by 5.4% in F1 score (one-to-one setting; Scenario 1). After fine-tuning, we evaluate the final model on the test set and report metrics including accuracy, precision (macro-averaged), recall, F1-score, AUROC (one-versus-rest), and AUPRC. The evaluation results of four scenarios are shown in Table 1 (main paper) and Tables 4-6.

**Table 7: Performance on downstream clustering.** Pre-training is performed on SLEEPEEG dataset, followed by an independent fine-tuning on EPILEPSY. We compare with five baselines including two non-transfer-based baselines (Random Init. and Non-DL), the best performing baseline in the context of classification task (i.e., TS-TCC), and two new models (TNC and CPC).

| Method | Silhouette | ARI | NMI |
|---|---|---|---|
| **Non-DL(KNN)** | 0.1208±0.0271 | 0.0549± 0.0059 | 0.0096± 0.0014 |
| **Random Init.** | 0.3497± 0.0509 | 0.3216±0.0136 | 0.2408± 0.0287 |
| **TNC** | 0.2353± 0.0018 | 0.0211± 0.0061 | 0.0082± 0.0018 |
| **CPC** | 0.2223±0.0011 | 0.0153± 0.0196 | 0.0063± 0.0055 |
| **TS-TCC** | 0.5154± 0.0458 | 0.6307± 0.0325 | 0.5178± 0.0283 |
| **TF-C (Ours)** | 0.5439± 0.0417 | 0.6583±0.0259 | 0.5567±0.0172 |

**Table 8: Performance on downstream anomaly detection.** We pre-train the model on FD-A and fine-tune on an anomaly detection subset of FD-B. The subset is highly imbalance (90% normal samples and 10% abnormal samples). We evaluate the performance with precision, recall, F1 score and AUROC.

| Models | Precision | Recall | F1 score | AUROC |
|---|---|---|---|---|
| **Non-DL(KNN)** | 0.4785±0.0356 | 0.6159±0.0585 | 0.5061±0.0278 | 0.7653±0.0153 |
| **Random Init.** | 0.8219±0.05319 | 0.7131±0.02773 | 0.7639±0.0094 | 0.8174±0.0218 |
| **TNC** | 0.8354±0.0484 | 0.7882±0.0157 | 0.7957±0.0165 | 0.8231±0.0379 |
| **CPC** | 0.5967±0.0776 | 0.4896±0.0866 | 0.5238±0.0792 | 0.7859±0.0356 |
| **TS-TCC** | 0.6219±0.0183 | 0.4431±0.0658 | 0.4759±0.0562 | 0.7966±0.0441 |
| **TF-C (Ours)** | 0.8526±0.0367 | 0.7823±0.0299 | 0.8312±0.0186 | 0.8598±0.0283 |

## Appendix G    Results on downstream clustering and anomaly detection tasks

Beyond the common downstream classification tasks, we conducted extensive experiments to examine the proposed TF-C in diverse downstream tasks. The comparison results with state-of-the-art baselines are shown in Table 7 for clustering and in Table 8 for anomaly detection. The experimental setups and results analysis are shown in Section 5.

## Appendix H    Further results on ablation study

We conduct ablation studies (Sec. 5) to evaluate the importance of every component in the developed TF-C model. For the example of one-to-one setting (**SLEEPEEG** → **GESTURE**) when pre-training the model using **SLEEPEEG** dataset and fine-tuning on **GESTURE** dataset, ablation study results are shown in Table 9. Through comparison, we observe that the full TF-C model achieves the highest performance in every evaluation metrics, indicating every component, especially the novel consistency loss $\mathcal{L}_C$), contributes to the model's performance.

## Appendix I    Visualization of embeddings in time-frequency space

To better demonstrate the effectiveness of the developed consistency loss $\mathcal{L}_C$, we visualize the learned embeddings in time-frequency space. In one-to-one setting, we pre-train the TF-C model on **SLEEPEEG** and fine-tune on **EPILEPSY** (Sec. 5.1). We visualize the testing samples of **EPILEPSY** in fine-tuning stage. The learned time-based embeddings $z_i^T$ and frequency-based embedding $z_i^F$ have 128 dimensions. We map the embeddings to a 2-dimensional space through UMAP (20 neighbors; minimum distance as 0.2; cosine distance) for visualization.

In this work, we develop a novel consistency loss $\mathcal{L}_C$ to induce the model learn closer time-based and frequency-based embeddings in a time-frequency space (Sec. 4.3). As our problem setting is that the fine-tuning dataset has small scale ($M \ll N$; Sec. 3), we randomly select 100 samples from the testing set (**EPILEPSY**) for visualization. The visualization of the learned embeddings in time-frequency space is shown in Figure 3 (without $\mathcal{L}_C$ v.s. with $\mathcal{L}_C$). We can observe that: (1) when without $\mathcal{L}_C$ (panel a), the time-based and frequency-based embeddings are clustered separately; the embeddings are largely clustered together when having $\mathcal{L}_C$ (panel b). The observation demonstrates that the proposed $\mathcal{L}_C$ has the ability to push time-based and frequency-based embeddings closer to each other. (2) for a specific sample $x_i$ (marked as dashed line), we can find that its time-based

**Table 9:** Ablation study (**SLEEPEEG** → **EPILEPSY**). If we remove $\mathcal{L}_{\mathrm{T}}$, we cannot calculate $\mathcal{L}_{\mathrm{C}}$, thus we remove both $\mathcal{L}_{\mathrm{T}}$ and $\mathcal{L}_{\mathrm{C}}$ (instead of only remove $\mathcal{L}_{\mathrm{T}}$). The same for $\mathcal{L}_{\mathrm{F}}$. "W/o $\mathcal{L}_{\mathrm{C}}$ and $\mathcal{L}_{\mathrm{T}}$" means removing the time encoder (Sec. 4.1) and the consistency module (Sec. 4.3), using $\boldsymbol{z}^{\mathrm{F}}$ as final embedding for downstream tasks in fine-tuning. "W/o $\mathcal{L}_{\mathrm{C}}$ and $\mathcal{L}_{\mathrm{F}}$" means removing the frequency encoder (Sec. 4.2) and the consistency module, using $\boldsymbol{z}^{\mathrm{T}}$ as final embedding for downstream tasks in fine-tuning. W/o $\mathcal{L}_{\mathrm{C}}$ means removing the consistency module, using $[\boldsymbol{z}^{\mathrm{T}}; \boldsymbol{z}^{\mathrm{F}}]$ as the final embedding for downstream tasks in fine-tuning. "Replace $\mathcal{L}_{\mathrm{C}}$ with $\mathcal{L}_{\mathrm{FF\text{-}C}}$" refers to replacing Eq. 3 by the distance $d(\boldsymbol{z}_i^{\mathrm{F}}, \widetilde{\boldsymbol{z}}_i^{\mathrm{F}}, \mathcal{D}^{\mathrm{pret}})$ between two frequency-based embeddings. "Replace $\mathcal{L}_{\mathrm{C}}$ with $\mathcal{L}_{\mathrm{TT\text{-}C}}$" refers to replacing Eq. 3 by the distance $d(\boldsymbol{z}_i^{\mathrm{T}}, \widetilde{\boldsymbol{z}}_i^{\mathrm{T}}, \mathcal{D}^{\mathrm{pret}})$ between two time-based embeddings.

| | Accuracy | Precision | Recall | F1 score | AUROC | AUPRC |
|---|---|---|---|---|---|---|
| W/o $\mathcal{L}_{\mathrm{C}}$ and $\mathcal{L}_{\mathrm{T}}$ | 0.7159±0.0128 | 0.7211±0.0428 | 0.7246±0.0428 | 0.7239±0.0429 | 0.8597±0.0236 | 0.7655±0.0386 |
| W/o $\mathcal{L}_{\mathrm{C}}$ and $\mathcal{L}_{\mathrm{F}}$ | 0.7327±0.0328 | 0.7246±0.0311 | 0.7339±0.0307 | 0.7317±0.0356 | 0.8991±0.0279 | 0.7236±0.0278 |
| W/o $\mathcal{L}_{\mathrm{C}}$ | 0.7428±0.0297 | 0.7289±0.0278 | 0.7451±0.0263 | 0.7377±0.0308 | 0.9125±0.0167 | 0.7706±0.0135 |
| Replace $\mathcal{L}_{\mathrm{C}}$ with $\mathcal{L}_{\mathrm{FF\text{-}C}}$ | 0.7259±0.0072 | 0.7319±0.0256 | 0.7338±0.0133 | 0.7341±0.0194 | 0.9015±0.0135 | 0.7529±0.0096 |
| Replace $\mathcal{L}_{\mathrm{C}}$ with $\mathcal{L}_{\mathrm{TT\text{-}C}}$ | 0.7124±0.0091 | 0.7256±0.0169 | 0.7231±0.0197 | 0.7296±0.0209 | 0.8726±0.0098 | 0.7627±0.0107 |
| Full Model (TF-C) | **0.7642±0.0196** | **0.7731±0.0355** | **0.7429±0.0268** | **0.7572±0.0311** | **0.9238±0.0159** | **0.7961±0.0109** |

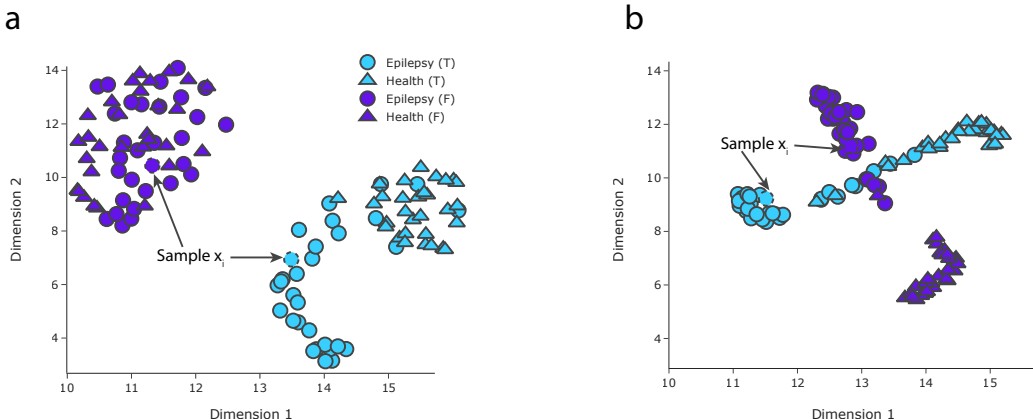

**Figure 3:** Visualizing time-based and frequency-based embeddings in **EPILEPSY** (one-to-one; pre-training on **SLEEPEEG**). (a) Without consistency loss item $\mathcal{L}_{\mathrm{C}}$ in both pre-training and fine-tuning. (b) With consistency loss item in pre-training and fine-tuning. Dark blue and light blue denote time-based embedding $\boldsymbol{z}_i^{\mathrm{T}}$ and frequency-based embedding $\boldsymbol{z}_i^{\mathrm{F}}$, respectively. Each circle denotes an 'epilepsy' sample, i.e., the subject has a seizure episode, while each triangle denotes a sample associated with 'health', i.e., the subject is healthy. The annotated $\boldsymbol{x}_i$ (marked as dashed line) is the same sample in two panels.

embedding (colored as light blue) and frequency-based (colored as dark blue) embedding are closer to each other in panel (b) than in panel (a). In particular, the cosine distance, between $\boldsymbol{z}_i^{\mathrm{T}}$ and $\boldsymbol{z}_i^{\mathrm{F}}$ of this sample, has reduced from 0.89 to 0.71 when using $\mathcal{L}_{\mathrm{C}}$ in our model. (3) the frequency-based embeddings of 'epilepsy' and 'health' samples are largely overlapped when without $\mathcal{L}_{\mathrm{C}}$ but clustered well when considering $\mathcal{L}_{\mathrm{C}}$, indicating the learned embeddings (with $\mathcal{L}_{\mathrm{C}}$) can potentially enhance downstream classification performance.

Furthermore, we attempt to quantify the impact of our consistency loss item on the quality of learned embeddings, using the one-to-many setting as shown in Figure 4. We pretrain the TF-C model on **SLEEPEEG** dataset and fine-tuning on **EPILEPSY**, **FD-B**, **GESTURE**, and **EMG**, respectively. Next, we explain how the cosine distance is calculated taking **EMG** dataset as an example. For every sample $\boldsymbol{x}_i$ in **EMG**, we measure the cosine distance between its time-based embedding $\boldsymbol{z}_i^{\mathrm{T}}$ and frequency-based embedding $\boldsymbol{z}_i^{\mathrm{F}}$ in the 128-dimension time-frequency space. Then, we take the average of the distance across all testing samples and report in Figure 4. The number of testing samples of each dataset can be found in Table 3. We find that the cosine distance is larger when removing the proposed consistency loss item $\mathcal{L}_{\mathrm{C}}$, which means our model, as designed and expected, indeed pushes the time-based and frequency-based embeddings closer to each other. For example, in **EPILEPSY**, the averaged cosine distance dropped 22.2% (from 1.32 to 1.08) when using consistency loss (Sec. 4.3). At the same time, the embeddings of samples drawn from different classes are more easily distinguished from each other, as shown in Figure 5: we observe that our TF-C model (with consistency loss $\mathcal{L}_{\mathrm{T}}$ increase the inter-class cosine distance from 0.88 to 1.42 with a remarkable

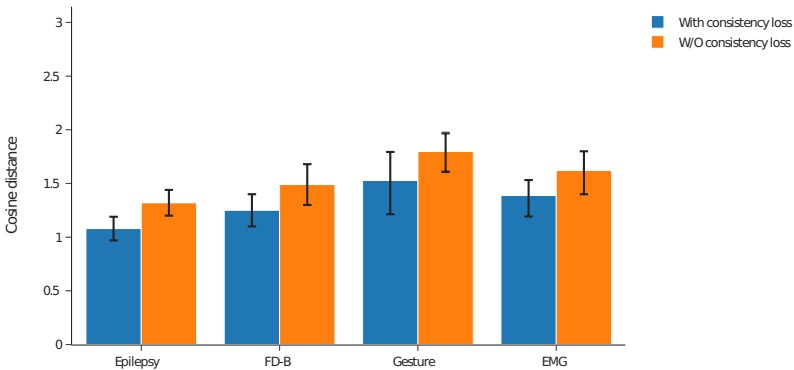

**Figure 4:** Cosine distance between *time-based and frequency-based embeddings*. Here, lower distance is better as it denotes higher consistency between time-based and frequency-based embeddings. The results are reported for the one-to-many setting, *i.e.*, pretraining the model on **SLEEPEEG** dataset and fine-tuning on **EPILEPSY**, **FD-B**, **GESTURE**, and **EMG**, respectively. In a specific dataset, *for each sample*, we measure the cosine distance between its time-based embedding $z_i^{\mathrm{T}}$ and frequency-based embedding $z_i^{\mathrm{F}}$ in a 128-dimension time-frequency space. In this figure, we report the cosine distance averaged across all testing samples. We observe that the cosine distance is larger when removing our proposed consistency loss item $\mathcal{L}_{\mathrm{C}}$, indicating our model indeed pushes the time-based and frequency-based embeddings closer to each other.

margin of 61.4% (one-to-many setting; testing set of **EPILEPSY**), indicating our model indeed has the ability to increase the distinctiveness of learned embeddings and bring forward better classification performance on the fine-tuning dataset of interest. The results of visualization increase our confidence that the developed TF-C pre-training model can capture generalizable properties and boost knowledge transfer across different time series datasets.

## Appendix J    Additional information on domain augmentations

To the best of our knowledge, this is the first work that direct perturbs the frequency domain of time series in contrastive learning. Here, we thoroughly explore various frequency augmentations and discuss the design of manipulation strategy in frequency domain. In particular, we analyze eight frequency augmentation policies and discuss their performance.

We consider three types of frequency augmentations:

- **Low- vs. High-band perturbations**: High-frequency components contribute to the fast varying parts of the time series (e.g., sharp transitions). In contrast, low-frequency components contribute to slow signal variations in the time domain. Low band perturbations correspond to the first half of the frequency spectrum.

- **Single- vs. Multi-component perturbations** (aka varying budget size): Perturbing several frequency components (i.e., larger budget) leads to more substantial changes in the time domain. In this analysis, we set the budget to 5 for multi-component perturbations and 1 for single-component perturbations.

- **Random vs. Distributional perturbations**: The selection of components in the spectrum that get perturbed is important because it influences the resulting time-based representation of the augmented sample. Frequency components can be sampled uniformly at random or according to a user-specified distribution (i.e., Gaussian in the following analysis).

Based on the above considerations, we developed eight frequency augmentations. In Fig. 6 (**EPILEPSY** dataset)), we visualize a series of perturbations in the frequency domain and their effects in the time domain (through the inverse FFT).

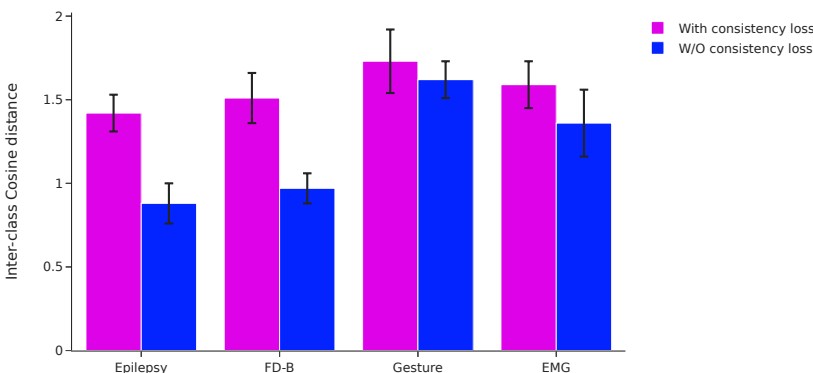

**Figure 5:** Cosine distance among *embeddings of samples belonging to different classes*. The higher distance indicates better discrimination and better downstream classification performance. As in Figure 4, the current experimental setup uses the one-to-many setting, but the calculation of cosine distance is slightly different. Take **EPILEPSY** as an example, we first measure the distance between a positive sample and all the negative samples, then take the averaged distance across all positive samples as inter-class distance. The distance is calculated based on the concatenation of time-based and frequency-based embeddings, *i.e.*, $z_i^{\text{tune}}$. We observe that our TF-C model (with consistency loss $\mathcal{L}_\text{C}$ increase the inter-class distance from 0.88 to 1.42 with a remarkable margin of 61.4% (one-to-many setting; testing set of **EPILEPSY**), indicating our model indeed has the ability to increase the distinction of learned embeddings and bring better classification performance on the fine-tuning dataset of interest. The margins in **GESTURE** and **EMG** are relatively low, and one potential reason is that the testing set of them are very small (120 for **GESTURE**; 41 for **EMG**).

In addition, we explore how different spectral perturbations affect the performance of TF-C. Table 10 shows results for the one-to-one setup (SLEEPEEG → EPILEPSY). We find the following:

- Perturbing low-band components performs better than augmenting high-band elements of the spectrum (0.8% change in F1 performance on average across the four setups). This can be explained by the nature of the underlying time series: in physiological signals, most information is carried in a low band (e.g., EEG signals are most informative in 0.5-70 Hz while > 70Hz is considered noise). This means that the choice of low- vs. high-band components depends on the signal context. For example, high-band components are informative for mechanical signals. We further verified this by external experiments in the FD-A → FD-B setting, where perturbing high-band components outperform low-band augmentations by 0.8% in F1. Thus, we consider perturbations across the entire spectral band in the paper to achieve broad generalization.

- In general, perturbing many frequency components degrades model performance. One reason for worse performance is that too big a budget incurs too significant changes in the time-based representation of augmented samples. In that case, the augmented samples would be easily distinguished by a contrastive model, leading to poor contrastive encoders. For this reason, we use single-component perturbations in the paper.

- We cannot draw a clear conclusion on whether the random selection of components is better than Gaussian distribution-based as both F1 scores are very similar. The optimal choice of sampling strategy likely depends on both the time-series signal and prediction task. We use random sampling in the paper.

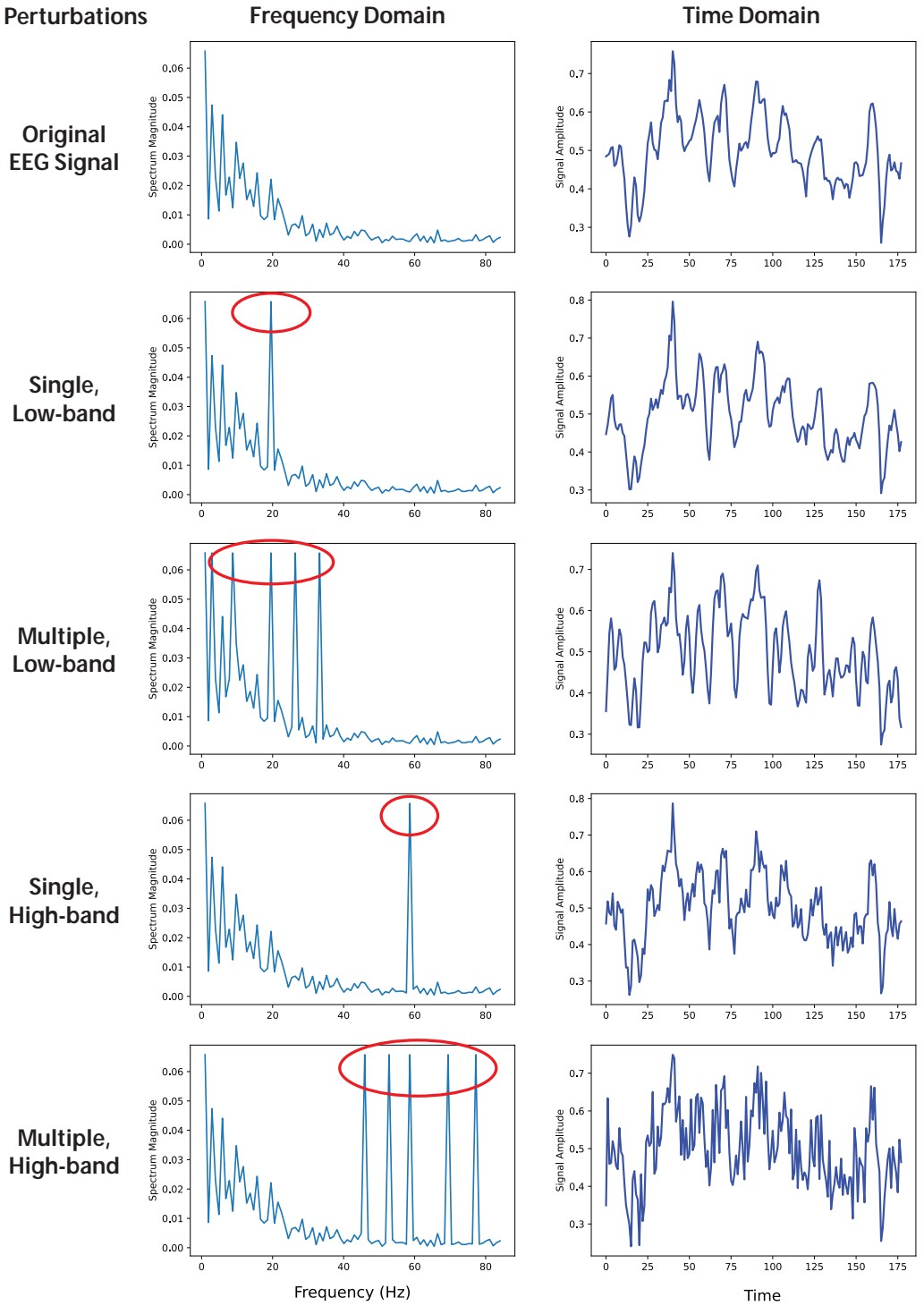

**Figure 6:** Illustration of augmentations in the frequency domain and the corresponding influence on the time domain. We considered the frequency augmentation from three aspects: low-band v.s. high-band, single-perturbation v.s. multiple perturbation, random selection v.s. distribution-based selection. The selection method has relative slight effect on time domain, thus, we mainly show how the perturbation budget (single or multiple) and frequency bands affect the temporal pattern in time domain. Here, for better visualization, we select $\alpha = 1$ which means the perturbed component has the maximum magnitude (we set $\alpha = 0.5$ in TF-C model) and set budget as 5 in multiple perturbation. The developed augmentations in this paper only perturb magnitude and keep phase information unchanged. We notice that perturbing phase is a crucial research direction in frequency domain augmentation in the future work.

**Table 10: Results comparison of eight frequency augmentations.** Considering the perturbing bands, the number of components, and the perturbation distribution, there are eight different combinations. Here we regard each factor as binary (e.g., Low v.s. High), we note that the more fine-grid factors (e.g., Delta band v.s. Alpha band v.s. Beta band v.s. Gamma band in the context of EEG signals) might be helpful for augmentation design.

| No. | Low vs. High | Single vs. Multi | Random vs. Gaussian | F1 score |
|-----|--------------|------------------|---------------------|----------|
| 1 | Low | Single | Random | 0.9136 |
| 2 | Low | Single | Gaussian | 0.8957 |
| 3 | Low | Multi | Random | 0.8763 |
| 4 | Low | Multi | Gaussian | 0.8859 |
| 5 | High | Single | Random | 0.8972 |
| 6 | High | Single | Gaussian | 0.8651 |
| 7 | High | Multi | Random | 0.8818 |
| 8 | High | Multi | Gaussian | 0.8775 |

## Appendix K  Discussion on many-to-one setting

We probed the one-to-one setting and one-to-many setting in knowledge transfer of time series (Sec. 5). Here, we discuss the many-to-one set up: pre-train on a mixture of multiple datasets and fine-tune on a single pure dataset.

We notice that the many-to-one setup fundamentally differs from the one-to-one and also from one-to-many configurations. Notably, in the one-to-one and one-to-many setups, the model is pre-trained on a single pre-training dataset presumed to originate from a data source exhibiting homogeneity. For example, samples in the pre-training dataset could describe similar devices (e.g., different kinds of boring machines) with similar sampling rates, lengths, semantic meanings, and/or relevant temporal patterns that occur on comparable time scales, etc. When some level of homogeneity exists in a pre-training dataset, the method can learn patterns and apply them to different fine-tuning datasets. In contrast, in many-to-one setups, pre-training datasets can be incredibly heterogeneous. For example, a pre-training dataset might contain samples describing boring machines' behavior and recordings of EEG medical devices. For this reason, pre-training can be severely hampered by diversity and misalignments across samples in the pre-training dataset. When we merge many heterogeneous samples into a single pre-training dataset, the model might struggle to find patterns underlying the combined dataset. While models in computer vision have been successfully pre-trained on large, diverse datasets, such as ImageNet, the literature on pre-training for time series is much sparser. Further, heterogeneous pre-training datasets pose additional challenges due to the unique properties of time series. Learning universal representations from multi-sourced time series datasets (i.e., many-to-one setups) remains an open challenge. It is an interesting future direction, and we hope this study can facilitate future research.

We have explored this and provide preliminary experimental results in four many-to-one setups (named N-to-one for N = 1,2, 3, and 4). Take four-to-one as an example. We merge four pre-training datasets (SLEEPEEG,FD-A, HAR, and ECG) into one pre-training dataset. We had to preprocess the datasets to make a many-to-one setup feasible. To address the mismatch between the number of channels, for the multivariate dataset (HAR), we take a single channel (i.e., x-axis acceleration) to make sure all samples have the same number of channels. To align the length across datasets, we make all samples have 1,500 observations by zero-padding (for the ones shorter than 1,500) and cutting-off (for the ones longer than 1,500). Further, we take 25% samples (instead of 100%; for computational reasons only) from each of the four datasets to create a pre-training dataset with 108,300 samples that belong to 18 classes in total. Similarly, we built pre-training for three-to-one (merge SLEEPEEG,FD-A, and HAR) and two-to-one (merge SLEEPEEG and FD-A).

We take the above N-to-one pre-training datasets and repeat the following analysis. First, we pre-train our TF-C and then fine-tune it on EPILEPSY for classification. The AUROC for one-to-one, two-to-one, three-to-one, and four-to-one settings are 0.9819, 0.9525, 0.8607, and 0.7253, respectively. We observe AUROC decreases with the increasing heterogeneity of pre-training datasets, which is consistent with our analysis in the first part of this answer. Nevertheless, whether there exists a framework that overcomes the problems and enables many-to-one pre-training for time series remains to be seen, and we believe it's an exciting direction for future research.

# Appendix References

[S1] Garrett Wilson and Diane J Cook. A survey of unsupervised deep domain adaptation. ACM Transactions on Intelligent Systems and Technology (TIST), 11(5):1–46, 2020.

[S2] Zhijun Zhou, Yingtian Zhang, Xiaojing Yu, Panlong Yang, Xiang-Yang Li, Jing Zhao, and Hao Zhou. Xhar: Deep domain adaptation for human activity recognition with smart devices. In 2020 17th Annual IEEE International Conference on Sensing, Communication, and Networking (SECON), pages 1–9. IEEE, 2020.

[S3] Felix Ott, David Rügamer, Lucas Heublein, Bernd Bischl, and Christopher Mutschler. Domain adaptation for time-series classification to mitigate covariate shift. arXiv:2204.03342, 2022.

[S4] Paulo Roberto de Oliveira da Costa, Alp Akçay, Yingqian Zhang, and Uzay Kaymak. Remaining useful lifetime prediction via deep domain adaptation. Reliability Engineering & System Safety, 195:106682, 2020.

[S5] Dezhi Hao and Xianwen Gao. Multi-weighted partial domain adaptation for sucker rod pump fault diagnosis using motor power data. Mathematics, 10(9):1519, 2022.

[S6] Penghao Zhang, Jiayue Li, Yining Wang, and Judong Pan. Domain adaptation for medical image segmentation: a meta-learning method. Journal of Imaging, 7(2):31, 2021.

[S7] Xiaoxia Wang, Haibo He, and Lusi Li. A hierarchical deep domain adaptation approach for fault diagnosis of power plant thermal system. IEEE Transactions on Industrial Informatics, 15(9):5139–5148, 2019.

[S8] Devis Tuia, Claudio Persello, and Lorenzo Bruzzone. Recent advances in domain adaptation for the classification of remote sensing data. arXiv:2104.07778, 2021.

[S9] Zhao-Hua Liu, Lin-Bo Jiang, Hua-Liang Wei, Lei Chen, and Xiao-Hua Li. Optimal transport-based deep domain adaptation approach for fault diagnosis of rotating machine. IEEE Transactions on Instrumentation and Measurement, 70:1–12, 2021.

[S10] Wei Lin, Anna Kukleva, Kunyang Sun, Horst Possegger, Hilde Kuehne, and Horst Bischof. Cycda: Unsupervised cycle domain adaptation from image to video. arXiv:2203.16244, 2022.

[S11] Dengyu Xiao, Chengjin Qin, Honggan Yu, Yixiang Huang, Chengliang Liu, and Jianwei Zhang. Unsupervised machine fault diagnosis for noisy domain adaptation using marginal denoising autoencoder based on acoustic signals. Measurement, 176:109186, 2021.

[S12] Youngeun Kim and Sungeun Hong. Adaptive graph adversarial networks for partial domain adaptation. IEEE Transactions on Circuits and Systems for Video Technology, 32(1):172–182, 2021.

[S13] Wei Zhang, Xiang Li, Hui Ma, Zhong Luo, and Xu Li. Universal domain adaptation in fault diagnostics with hybrid weighted deep adversarial learning. IEEE Transactions on Industrial Informatics, 17(12):7957–7967, 2021.

[S14] Ronghang Zhu, Xiaodong Jiang, Jiasen Lu, and Sheng Li. Cross-domain graph convolutions for adversarial unsupervised domain adaptation. IEEE Transactions on Neural Networks and Learning Systems, 2021.

[S15] Zhengyang Chen, Shuai Wang, and Yanmin Qian. Self-supervised learning based domain adaptation for robust speaker verification. In ICASSP 2021-2021 IEEE International Conference on Acoustics, Speech and Signal Processing (ICASSP), pages 5834–5838. IEEE, 2021.

[S16] Kazuma Fujii and Kazuhiko Kawamoto. Generative and self-supervised domain adaptation for one-stage object detection. Array, 11:100071, 2021.

[S17] Hao Guan and Mingxia Liu. Domain adaptation for medical image analysis: a survey. IEEE Transactions on Biomedical Engineering, 2021.

[S18] Wentao Mao, Ling Ding, Yamin Liu, Sajad Saraygord Afshari, and Xihui Liang. A new deep domain adaptation method with joint adversarial training for online detection of bearing early fault. ISA transactions, 122:444–458, 2022.

[S19] Haifeng Xia, Handong Zhao, and Zhengming Ding. Adaptive adversarial network for source-free domain adaptation. In Proceedings of the IEEE/CVF International Conference on Computer Vision, pages 9010–9019, 2021.

[S20] Muhammad Awais, Fengwei Zhou, Hang Xu, Lanqing Hong, Ping Luo, Sung-Ho Bae, and Zhenguo Li. Adversarial robustness for unsupervised domain adaptation. In Proceedings of the IEEE/CVF International Conference on Computer Vision, pages 8568–8577, 2021.

[S21] Yuxin Ma, Yang Hua, Hanming Deng, Tao Song, Hao Wang, Zhengui Xue, Heng Cao, Ruhui Ma, and Haibing Guan. Self-supervised vessel segmentation via adversarial learning. In Proceedings of the IEEE/CVF International Conference on Computer Vision, pages 7536–7545, 2021.

[S22] Qiao Liu and Hui Xue. Adversarial spectral kernel matching for unsupervised time series domain adaptation. In Int Joint Conf Artif Intell, pages 2744–2750, 2021.

[S23] Yuan Yuan and Lei Lin. Self-supervised pretraining of transformers for satellite image time series classification. IEEE Journal of Selected Topics in Applied Earth Observations and Remote Sensing, 14:474–487, 2020.

[S24] Colorado J Reed, Xiangyu Yue, Ani Nrusimha, Sayna Ebrahimi, Vivek Vijaykumar, Richard Mao, Bo Li, Shanghang Zhang, Devin Guillory, Sean Metzger, et al. Self-supervised pretraining improves self-supervised pretraining. In Proceedings of the IEEE/CVF Winter Conference on Applications of Computer Vision, pages 2584–2594, 2022.

[S25] Tianlong Chen, Sijia Liu, Shiyu Chang, Yu Cheng, Lisa Amini, and Zhangyang Wang. Adversarial robustness: From self-supervised pre-training to fine-tuning. In Proceedings of the IEEE/CVF Conference on Computer Vision and Pattern Recognition, pages 699–708, 2020.

[S26] Priya Goyal, Mathilde Caron, Benjamin Lefaudeux, Min Xu, Pengchao Wang, Vivek Pai, Mannat Singh, Vitaliy Liptchinsky, Ishan Misra, Armand Joulin, et al. Self-supervised pretraining of visual features in the wild. arXiv:2103.01988, 2021.

[S27] Peri Akiva, Matthew Purri, Kristin Dana, Beth Tellman, and Tyler Anderson. H2o-net: Self-supervised flood segmentation via adversarial domain adaptation and label refinement. In Proceedings of the IEEE/CVF Winter Conference on Applications of Computer Vision, pages 111–122, 2021.

[S28] Xinyi Wu, Zhenyao Wu, Hao Guo, Lili Ju, and Song Wang. Dannet: A one-stage domain adaptation network for unsupervised nighttime semantic segmentation. In Proceedings of the IEEE/CVF Conference on Computer Vision and Pattern Recognition, pages 15769–15778, 2021.

[S29] Alexei Baevski, Michael Auli, and Abdelrahman Mohamed. Effectiveness of self-supervised pre-training for speech recognition. arXiv:1911.03912, 2019.

[S30] Matthew McDermott, Brenda Yap, Peter Szolovits, and Marinka Zitnik. Structure inducing pre-training. arXiv:2103.10334, 2021.

[S31] W Hu, B Liu, J Gomes, M Zitnik, P Liang, V Pande, and J Leskovec. Strategies for pre-training graph neural networks. In International Conference on Learning Representations (ICLR), 2020.