# OpenReview forum: "Self-Supervised Contrastive Pre-Training For Time Series via Time-Frequency Consistency"
_NeurIPS.cc/2022/Conference — NeurIPS 2022 Accept_

### Official Review · Reviewer_UjqE · 2022-07-03

**Rating:** 8
**Confidence:** 4
**Soundness:** 3 good
**Presentation:** 3 good
**Contribution:** 3 good

**Summary:**

This paper proposes a new method for self-supervised pre-training for time series datasets. These pre-trained models can then be fine tuned for classification on different downstream time series datasets.

In addition to using augmentations on time series data in a contrastive learning setting to obtain time series specific  representations , the authors also input frequency domain (FFT)  for input time series. These frequency domain inputs  are  also augmented through frequency domain perturbations. These frequency domain inputs and their corresponding perturbations are then used by a contrastive loss to learn frequency domain representations.

These time series and frequency domain representations are then projected to a  Time-freqeuncy space where the the time series representations and their corresponding frequency domain representations are encouraged to be closer than the time series representations and its corresponding augmented frequency domain representations (or frequency domain and its augmented time series representation) through a margin loss.

This methodology results in representations ,which when fine tuned on different types of time series datasets, leads to improved classification performance over other self-supervised representation learning baselines.

**Questions:**

Could the authors provide more information about the permutation budget needed for frequency domain perturbations ?
 The authors mention that the perturbation budget is to ensure that the perturbations do not change the input seignifcantly but more details on this could be helpful (as  some low frequency pertbuations might change the signal more than high frequency perturbations , though this might depend on the input sequence ).
Some details on how these perturbations affect the learned representations could be helpful for the readers.



**Limitations:**

The authors have talked about some limitations in their work (e.g how the method can't directly work for irregulary sampled data but encoder based methods could help solve this)


Also I think mentioning how different time series provide different types of representations which are more suitable for certain downstream time series classification datasets (like human activity pre train and fine tune gesture) could be helpful when discussing the limitations.
(The authors do mention in the results section, but mentioning in the limitations could be helpful )

Also  would this method work for really high dimensional datasets ? (Like 100 dimensional?)


**Strengths And Weaknesses:**

Pros
- The paper proposes a novel direction for self-superivison for time series datasets through both, perturbations in the frequency domain , as well as a join time-frequency representation space
- There has been relatively little work on learning self-supervised representations for time series datasets (at least as compared to image/vision/nlp domains). This work addresses this and presents a method which could be useful for the community.
- The method was extensively evaluated on different types of downstream time series classification tasks (that range from ECG, to human activity recognition and motor fault detection) and showed promising results
- The method was mostly  clearly described and the paper was was to follow.

Cons
(This isn't really  a con,
- I had some issues with section 3, particularly the paragraph titled "Rationale for Time-Frequency Consistency" where some of the sentences were a bit confusing. E.g line 162. "the relationship between them provides an invariance" that can be exploited. I understand how the relationship between  time domain and frequency domain could be invariant , but I am having difficulty of seeing how this could be useful for self-supervised representations. I am thinking of this interns  invariant representations learned for images, where an image and its rotated form should have invariant representations, and is more intuitive to understand.
  Maybe more details on how the time-frequency relationship invariant can be useful for self-supervision could be helpful (perhaps though a simple example)

---

> ### Author Response · Authors · 2022-08-02
> **Response to Reviewer UjqE Part I**
>
> We are grateful for your thoughtful feedback and appreciate your comments on the strengths and possible areas for improvement. We are delighted that you find our ideas easy to follow, this research direction novel, and experimental evaluations extensive. The following response addresses the questions about our method’s rationale, implementation, and limitations. Thank you very much for your valuable perspective!
>
>
> **Q1**: I had some issues with section 3, particularly the "Rationale for Time-Frequency Consistency." I understand how the relationship between the time and frequency domains could be invariant, but I am having difficulty seeing how this could be useful for self-supervised representations. I am thinking of these invariant representations learned for images, where an image and its rotated form should have invariant representations and is more intuitive to understand.
>
> **A1**: We greatly appreciate the suggestions and are confident they will help improve the paper. Our response provides a simple example based on an analogy from computer vision.
>
> In computer vision, it is well established that an image and its variants obtained by simple transformations (e.g., rotation, translation, scaling, etc.) should have invariant representations. These invariances are used as augmentations to guide the learning of self-supervised representations. The intuition behind these and other transformations is that the underlying object in the image does not change no matter how the image is rotated, translated, re-scaled, etc. In other words, information carried by the original image and the transformed image is the same.
>
> Similarly, the time domain and frequency domain representations carry the same information in time series. Thus it is reasonable to expect that by exploring local neighborhoods in the time domain and frequency domain and enforcing consistency of feature representations inter- and intra- domains, invariance properties are captured in our model via self-supervised learning. Thus,* information carried in time and frequency domains of the same or similar time series sample should be the same*. This invariance, formalized as “Time-Frequency Consistency,” is helpful for self-supervised pre-training. We will include the above discussion in the camera-ready version.

---

> > ### Comment · Reviewer_UjqE · 2022-08-09
> > **Response to comments**
> >
> > I would like to thank the authors for their detailed response to my questions. I think most of my questions have been answered. As the authors mentioned, adding some of the details in your comments to the main paper would help strengthen an already strong paper.
> >
> > Particularly, the added details on intuition behind time-frequency consistency in your comments helped my understanding and I believe the readers would also find this useful.
> >
> > Additional details/experiments  on the different frequency perturbations were also helpful!

---

> ### Author Response · Authors · 2022-08-02
> **Response to Reviewer UjqE Part II**
>
> **Q2**: Could the authors provide more information about the permutation budget needed for frequency domain perturbations? The authors mention that the perturbation budget ensures that the perturbations do not change the input significantly. Still, more details on this could be helpful (as some low-frequency perturbations might change the signal more than high-frequency perturbations, though this might depend on the input sequence).
>
>
> **A2**: We agree that adding clarifications on this would be insightful. We have thoroughly explored various frequency augmentations. We are sorry the paper did not discuss all of them, and we will work on clarifying them. To address this, we here analyze eight frequency augmentation policies and discuss their performance.
>
> We consider three types of frequency augmentations:
>
> - *Low- vs. High-band perturbations:* High-frequency components contribute to the fast varying parts of the time series (e.g., sharp transitions). In contrast, low-frequency components contribute to slow signal variations in the time domain. Low band perturbations correspond to the first half of the frequency spectrum.
>
> - *Single- vs. Multi-component perturbations (aka varying budget size):* Perturbing several frequency components (i.e., larger budget) leads to more substantial changes in the time domain. In this analysis, we set the budget to 5 for multi-component perturbations and 1 for single-component perturbations.
>
> - *Random vs. Distributional perturbations:* The selection of components in the spectrum that get perturbed is important because it influences the resulting time-based representation of the augmented sample. Frequency components can be sampled uniformly at random or according to a user-specified distribution (i.e., Gaussian in the following analysis).
>
> Based on the above considerations, we developed eight frequency augmentations. To visualize perturbations in the frequency domain and their effects in the time domain (through the inverse FFT), we include a new figure in the paper (Fig. 6 with ten panels; Epilepsy dataset; see revised Appendix).
>
> In addition, we conducted new experiments to explore how different spectral perturbations affect TF-C performance. The following table shows results for the one-to-one setup (SleepEEG $\rightarrow$ Epilepsy). We find the following:
>
> - Perturbing low-band components performs better than augmenting high-band elements of the spectrum (0.8% change in F1 performance on average across the four setups). This can be explained by the nature of the underlying time series—in physiological signals, most information is carried in a low band (e.g., EEG signals are most informative in 0.5-70 Hz while > 70Hz is considered noise). This means that the choice of low- vs. high-band components depends on the signal context. For example, high-band components are informative for mechanical signals. We further verified this by external experiments in the FD-A $\rightarrow$ FD-B setting, where perturbing high-band components outperform low-band augmentations by 0.8% in F1. Thus, we consider perturbations across the entire spectral band in the paper to achieve broad generalization.
>
> - In general, perturbing many frequency components degrades model performance. One reason for worse performance is that too big a budget incurs too significant changes in the time-based representation of augmented samples. In that case, the augmented samples would be easily distinguished by a contrastive model, leading to poor contrastive encoders. For this reason, we use single-component perturbations in the paper.
>
> - We cannot draw a clear conclusion on whether the random selection of components is better than Gaussian distribution-based as both F1 scores are very similar. The optimal choice of sampling strategy likely depends on both the time-series signal and prediction task. We use random sampling in the paper.
>
> We will include the above analyses, experimental results, and insights in the camera-ready version. In addition, we note that the developed augmentations in this paper perturb magnitude and keep phase information unchanged. The phase of perturbation is a crucial direction in frequency domain augmentation in future work. Finally, we agree that frequency augmentations represent a fruitful direction for future research, which we will address in the limitation and future work section.
>
>
> | Frequency augmentations | Low vs. High | Single vs. Multi | Random vs. Gaussian | F-1 score |
> |-------|--------------|--------------|--------------|--------------|
> | 1| Low  | Single | Random | 0.9136 |
> | 2| Low  | Single | Gaussian| 0.8957 |
> | 3| Low  | Multi | Random | 0.8763 |
> | 4| Low  |  Multi | Gaussian| 0.8859|
> | 5| High | Single | Random | 0.8972 |
> | 6| High  | Single | Gaussian| 0.8651 |
> | 7| High  |  Multi | Random | 0.8818 |
> | 8| High  |  Multi | Gaussian| 0.8775 |

---

> ### Author Response · Authors · 2022-08-02
> **Response to Reviewer UjqE Part III**
>
> **Q3**: Would this method work for really high-dimensional (e.g., 100 dimensions) datasets?
>
> **A3**: Thank you for raising this point. High-dimensional time series are unlike other input types found in deep learning literature. Most ML methods assume that samples are IID, which is not the case for most time series. Additional difficulties arise in the presence of epistemic uncertainty or nonstationary inputs [R4]. If that is the case, further signal processing must be done before training, and/or the ML method must change from batch to online. However, some techniques can help perform unsupervised and supervised learning tasks when time series are high-dimensional. Standard practices include reducing the high dimensionality of the input and using a suitable embedding for sequences so that the LSTM or transformer can handle time dependencies [R5].
>
> The TF-C method can handle univariate and moderately sized multivariate time series. However, one needs to exploit inter-sensor relationships to apply TF-C for time series with a massive number of sensors/channels/dimensions. For example, to deal with high-dimensional time series, one can model pairwise interactions between sensors using sensor dependency graphs (as in Raindrop [68]) and leverage the learned inter-sensor dependencies for fine-tuning involving an arbitrary number of sensors. We will expand the discussion on high-dimensional time series in the camera-ready version.
>
> [68] Zhang X, Zeman M, Tsiligkaridis T, et al. Graph-Guided Network for Irregularly Sampled Multivariate Time Series. ICLR. 2022.
>
> [R4] Kuznetsov V, Mohri M. Time series prediction and online learning. In Conference on Learning Theory, 2016.
>
> [R5] Ayesha S, Hanif MK, Talib R. Overview and comparative study of dimensionality reduction techniques for high dimensional data. Information Fusion, 2020.

---

### Official Review · Reviewer_miad · 2022-07-04

**Rating:** 8
**Confidence:** 4
**Soundness:** 4 excellent
**Presentation:** 4 excellent
**Contribution:** 3 good

**Summary:**

The authors propose a new way of training self-supervised models for time series, such that time-frequency consistency is enforced. To this end, the authors add two additional loss terms to an earlier-introduced contrastive loss. The two additional terms enforce time, respectively frequency consistency, inspired by the triplet loss. Performance of the proposed model is compared to several strong baselines, and the proposed model is shown to outperform most baselines on most metrics.

**Questions:**

Could the authors explain how it is possible that the random initialization method for one-to-many in sleepEEG --> Fd-B performs almost as good as the proposed model, and much better than most of the baselines on the AUPRC metric?

How sensitive is the performance to the setting of lambda?

Two additional minor things:
- Fig 1: the symbols for the embeddings of the time-encoder have an F superscript instead of a T.
- L277: "have" should be "has"


**Limitations:**

Yes, limitations were discussed.

**Strengths And Weaknesses:**

 Originality: The proposed approach is simple, but elegant.

Quality: The paper is of good quality.

Clarity:  The paper is well-written and easy to follow, and the figures are also clear and informative, therewith helping the reader to gain a better understanding.

Significance: The field of self-supervised learning has so far mostly been focused on imaging tasks, while many real life data are time series. Development in self-supervised learning methods, dedicated to time series is, therefore, highly significant.

---

> ### Author Response · Authors · 2022-08-02
> **Response to Reviewer miad**
>
> Thank you for your very thoughtful feedback. We highly appreciate that you recognized the novelty and significance of this work. We are also glad that you have found the paper easy to follow and the figures informative for understanding our problem-solving approach. Again, we thank you for your comments and would love to hear any follow-up you have to this response.
>
> **Q1**: How can the random initialization method perform almost as well as TF-C for one-to-many in SleepEEG -->Fd-B and much better than most baselines on the AUPRC metric?
>
> **A1**: Thanks for bringing up this question. We appreciate your very keen observation. Our response contains two parts.
>
> We start by explaining the performance of Random Init. on one-to-many setups (SleepEEG $\rightarrow$ FD-B) shown in the second block of Table 2. In absolute points, TF-C outperforms Random Init. in 22% (Accuracy), 27% (Precision), 20% (Recall), 25% (F1 score), 11% (AUROC), and 3% (AUPRC). Although Random Init. has competitive performance with TF-C in AUPRC, it performs poorly on the other five evaluation metrics. One potential reason is that Random Init. can obtain a relatively good classification probability near the decision boundary (which gives it high AUROC and AUPRC values) but not high enough to cross the boundary and flip predicted labels, resulting in low Accuracy, Precision, Recall, and F1 scores.
>
> Second, regarding the comparison between Random Init. and other baselines, we observe that most baselines outperform Random Init. in Accuracy, Precision, Recall, and F1 score. Random Init. indeed achieves better performance than TS2Vec and SimCLR. We hypothesize that TS2Vec and SimCLR cannot handle the knowledge transfer from the physiological signal (EEG) to the mechanical signal (vibration), which is challenging because of the underlying domain shift. If that is the case, poor initialization brought by pre-training degrades performance on fine-tuning, producing worse results than using random initialization alone, which is known in the literature as a negative transfer problem. Negative transfer is pervasive and particularly challenging in time series (see Figs. 4 and 5 in [16], where most transfers are negative). This challenge has motivated our generalizable property for time series and has led to Representational Time-Frequency Consistency (TF-C) and the development of the TF-C method.
>
> We will include the above explanation and discussion in the camera-ready version. We really appreciate the clarifications and the chance to engage in debate about this work.
>
> [16] Fawaz H I, Forestier G, Weber J, et al. Transfer learning for time series classification, 2018 IEEE Big Data, 2018.
>
> **Q2**: How sensitive is the performance to the setting of lambda?
>
> **A2**: We performed new experiments to investigate the influence of the $\lambda$ coefficient in Eq. (4). The following table shows results for the one-to-one setting (HAR $\rightarrow$ Gesture), where we varied the coefficient between 0 and 1. We find that too small or too large $\lambda$ can lead to worse performance: small $\lambda$ emphasizes the TF-C property, whereas large $\lambda$ emphasizes individual time/frequency contrastive encoders (as shown in Eq (4) in the initial submission). Therefore, the $\lambda = 0.5$ is used in our experiments to balance individual time/frequency losses and the consistency loss.
>
> | $\lambda$ | 0 |0.1 | 0.3 | 0.5 |0.7 |0.9| 1.0|
> | ---|----|---|----|---|---|----|----|
> |F-1 score | 0.7541 | 0.7883 | 0.7987 | 0.7991 | 0.7824 | 0.7869| 0.7436|
>
>
> We are grateful for your valuable perspective and for recognizing our work!

---

> > ### Comment · Reviewer_miad · 2022-08-08
> > **Response to authors**
> >
> > Thank you for the response.
> > Including the presented additions in the camera ready version will improve the paper.
> > I'm still happy with the paper and it deserves an accept.

---

### Official Review · Reviewer_hNJe · 2022-07-11

**Rating:** 6
**Confidence:** 5
**Soundness:** 3 good
**Presentation:** 3 good
**Contribution:** 3 good

**Summary:**

This paper proposes a self-supervised contrastive pre-training target for time series via time-frequency consistency. Based on the assumption that time-based embedding and frequency-based embedding for a particular time series example should be close together in time-frequency space, consistency loss is introduced. Evaluation of eight real-world time series datasets under one-to-one and one-to-many transfer settings, the proposed pre-training target is proved to have better performance over existing methods.

=====================

Updates in the rebuttal period: Thanks for the authors' responses about updated clarifications and extended experiments. I have read them carefully and raised my scores.

**Questions:**

1. What are the detailed augmentation and frequency-domain transformation settings in the experiments? How are different augmentations compared in the frequency domain?


**Limitations:**

1. The novelty should be discussed a step further.
2. Theoretical analysis, further experiments, or case studies about frequency domain augmentations should be added.
3. More baselines should be compared.

**Strengths And Weaknesses:**

Strength:
1. The paper is clearly written and well presented.
2. Experiments on eight real-world datasets are comprehensive.

Weakness:
1. The novelty should be discussed a step further. This paper claims that "this is the first work that develops augmentations in frequency domain", however, this sentence is ambiguous. Does this paper first "augment time series self-supervised learning in frequency domain" or "develops frequency domain augmentations"? This should be awared because the reviewer has noticed that there is a relevant method BTSF to appear in ICML 2022 (first appeared in arxiv in 8 Feb 2022), using a close temporal-spectral fusion idea for time-series representation learning, which involves both time domain and frequency domain for contrastive learning as well (amazingly experimental SLEEPEEG, EPILEPSY, HAR datasets in this paper are also used in BTSF). Please consider rephrasing this claim based on the above fact. Besides, the differences and specific novel ideas of this paper to that of BTSF should be discussed.
2. More baselines should be compared. Many baselines listed in the related work are supposed to be compared in the experiments as well. Especially some of them show better performance than the currently selected baselines. Besides, please consider update the references, for example, [45]: arxiv -> ICLR 2022, [46]: arxiv -> AAAI 2022, [47]: arxiv -> IJCAI 2021.
3. The experiments only involves classification tasks without regression task, forecasting task, anomally detection task. Hence, it is suggested to point out that this paper aims at this specific task in the title and introduction.
4. Further understanding of frequency augmentation is needed. As a fundamental concept and assumption of the paper, the time-frequency consistency's underlying mechanism is not discussed in this paper. This paper verified its effectiveness, but doesn't provide further analysis about how it works. Theoretical analyses are also missing. Besides, as mentioned in section 4.2, augentation in frequency domain may change the time series significantly in time domain. Does the augmented time series still make sence? Since no case study is given, the effectiveness and logic of this operation is questionable.
5. Minors: - "For" -> "for" in the title.

---

> ### Author Response · Authors · 2022-08-02
> **Response to Reviewer hNJe Part I**
>
> We are happy that you felt our paper is clearly written, well presented, and reports comprehensive experiments. Thank you very much for your valuable perspective! Below, we respond in detail to each of your concerns. If, after reading our responses, you do not feel we have sufficiently justified a higher score, please let us know where we can further improve our work or what concerns you still have. Thank you again!
>
>
> **Q1**: The novelty should be discussed a step further. This paper claims that "this is the first work that develops augmentations in the frequency domain.” However, this sentence is ambiguous. This should be addressed because the reviewer noticed a related method, BTSF, to appear in ICML 2022 (first appeared on arXiv on 8 Feb 2022). Consider rephrasing this claim based on the above fact. Besides, the differences and novel ideas of this paper to that of BTSF should be discussed.
>
> **A1**:  Thanks for bringing up this point. We agree entirely that clarifying the novelty and significance of our paper’s contributions is essential. Our response to this has three parts.
>
> (1) First, thanks for bringing up the BTSF manuscript. At the time of the NeurIPS 2022 submission deadline (May 16, 2022), the BTSF method was not published. It was described in a non-published preprint. Further, no code and no datasets were available to reproduce the results. Additionally, we hope that it is obvious that we could not know the BTSF manuscript was under review at ICML 2022. Although it appeared on arXiv in February 2022, it was not published [R1] until after the NeurIPS’22 submission deadline.
>
> After carefully reading the ICML 2022 publication of the BTSF method (https://proceedings.mlr.press/v162/yang22e.html), we cannot find the code or open-source implementation of the method to include it in our experiments and reproduce BTSF results. Therefore, we clarify that our work is different from BTSF in the following key aspects:
>
> - Problem definitions for both papers are different. Our method is designed to produce generalizable representations that can transfer to a different time series dataset (going from pre-training to a fine-tuning dataset) for the purpose of *transfer learning*. In contrast, BTSF attempts to learn embeddings within the same dataset for the purpose of *representation learning*. Our model captures the TF-C property invariant to different time series (in terms of various temporal dynamics, semantic meaning, etc.) and can thus serve as a vehicle for transfer learning. In contrast, BTSF learns embeddings invariant to perturbations (i.e., instance-level dropout) of the same time series.
>
> - The modeling of the frequency domain is different in both papers. We developed augmentations in the frequency domain based on the spectral properties of time series. In contrast, although BTSF involves a frequency domain, *its data transformation is solely implemented in the time domain* (using instance-level dropout; Sec. 3.1 in BTSF). That is, the BTSF method applies the FFT after augmenting samples in the time domain which can lead to information loss.
>
> - BTSF emphasizes fusing temporal and spectral features to generate discriminative embeddings. Unlike BTSF, our model leverages the consistency between time-based and frequency-based embeddings to produce generalizable time series representations. Our model maps every sample to a time-frequency embedding space and constrains the relative relationships between embeddings (through triplet loss, Sec. 4.3 in TF-C) according to the TF-C property. The underlying consistency allows TF-C to realize transfer learning across time series datasets, which is TF-C’s unique advantage.
>
> (2) Second, we agree it is important to rephrase the claim in our paper based on the above discussion. TF-C introduces frequency domain augmentations in the sense that it directly perturbs the frequency spectrum. TF-C is the first method that uses frequency domain augmentations to enable transfer learning in time series.
>
> (3) Third, in response to the reviewer’s comment, we made the following changes to the paper. i) We clarified the novelty of frequency augmentations (updated Introduction and Related Work). ii) We cited BTSF and discussed our specific novelty compared to it (updated Related Work). iii) We analyzed types of frequency augmentations in the frequency domain and added experiments to analyze their influence on performance (see our answers to your Q4).
>
> [R1] Yang L, Hong S. Unsupervised Time-Series Representation Learning with Iterative Bilinear Temporal-Spectral Fusion. ICML. 2022: 25038-25054.

---

> ### Author Response · Authors · 2022-08-02
> **Response to Reviewer hNJe Part II**
>
> **Q2**: More baselines should be compared. The experiments should also compare many baselines listed in the related work. Consider updating the references, for example, [45]: arxiv -> ICLR 2022, [46]: arxiv ->AAAI 2022, [47]: arxiv -> IJCAI 2021.
>
>
> **A2**: Our response to concerns around baselines has four parts:
>
> (1) In new experiments, we compare TF-C to two state-of-the-art methods listed in the Related Work section, TNC [45] and CPC [29]. Note that the BTSF method mentioned in Q1 does not release public code (neither on arXiv nor in the published ICML version). Therefore, we had to exclude the BTSF method from experiments as we didn’t have enough time to faithfully reimplement it from scratch in a short rebuttal period. While TNC and CPC are designed for representation learning tasks (instead of pre-training, which is the focus of our study), we adopted them for pre-training and kept fine-tuning settings the same as in TF-C to make all results comparable.
>
> (2) We used TNC and CPC implementations provided by the authors of the TNC paper, available at https://github.com/sanatonek/TNC_representation_learning, and followed the original authors’ recommendations on hyperparameters. The following table shows results for the one-to-one setting where we pre-trained each method on SleepEEG and fine-tuned it for Epilepsy prediction. Results show that, on average, TF-C outperforms TNC and CPC in Accuracy by 23% (absolute value). We hypothesize that TNC did not perform well because it is specifically developed for long time-series samples. This hypothesis is further supported by our finding that TNC achieved competitive performance on the long-sequence FD-B dataset (length: 5120 time stamps), as shown in the table displaying results for anomaly detection (see our answer to your Q3). In comparison, CPC uses the predictive coding principle to learn observation-level representations through an autoregressive model. Furthermore, an inspection of the waveform of the SleepEEG samples shows that the signals are highly non-stationary, which may impede effective prediction in the case of CPC. Taken together, these new experiments boost our confidence in the effectiveness of TF-C.
>
> | Method   | Accuracy | Precision | Recall | F1 score |  AUROC | AUPRC |
> |-------|--------------|--------------|--------------|--------------|--------------|--------------|
> | TNC  | 0.6701+/-0.0071 |0.5336 +/- 0.2742|0.5011 +/- 0.0016 |0.4024 +/- 0.0033 |0.5853 	+/- 0.0487 |0.6276 +/- 0.0438|
> | CPC  | 0.662	+/- 0.011|0.434 +/- 0.225| 0.5028 +/- 0.0064 |0.406 	+/- 0.0134|0.5722+/-0.0748 | 0.6145 +/- 0.0575|
> | TF-C  |0.9495+/-0.0249 | 0.9456+/-0.0108| 0.8908+/-0.0216| 0.9149+/-0.0534| 0.9811+/-0.0237 |0.9703+/-0.0199|
>
> (3) We will include these additional experiments in the camera-ready version. *Together with the new baselines, our paper now has a total of 10 baselines.* Results for the other eight baselines (Non-DL(KNN), Random Init., TS-SD, TS2vec, CLOCS, Mixing-up, TS-TCC, and SimCLR) are in Table 1 in our initial submission. We sincerely thank the reviewer for pointing this out and would love to hear any follow-up you have to this response.
>
> (4) Finally, we carefully checked the reference lists and updated ten citations from arXiv to published versions.

---

> ### Author Response · Authors · 2022-08-02
> **Response to Reviewer hNJe Part III**
>
> **Q3**: The experiments only involve classification tasks without regression, forecasting, and anomaly detection. Point out that this paper aims at this specific task in the title and introduction.
>
> **A3**: Thank you for pointing out this area that we should clarify. We apologize if the prediction task was unclear in the paper's original presentation and will work hard to clarify it so this message is better conveyed. Our response has three parts:
>
> First,  while we focus on knowledge transfer across diverse time series datasets, we agree that the current embedding scheme and loss functions are more suitable to capture global representations of time series. Our model was designed to capture the generalizable pattern aiming at knowledge transfer, which is favorable for classification tasks that leverage global information over tasks that use local context (e.g., forecasting). As you suggested in the review, in the camera-ready version, we will update our manuscript thoroughly (e.g., title, abstract, introduction, methods) to clarify that this work is designed mainly for time series classification.
>
> Second, beyond the above point, we did new experiments to demonstrate how our approach applies to clustering and anomaly detection. We compared our TF-C with 5 baselines on these tasks: 2 non-transfer-based baselines (Random Init. and Non-DL), the best performing baseline currently included in the paper (i.e., TS-TCC), and 2 new methods included just for this experiment (TNC and CPC).
>
> - **Clustering:** In the one-to-one setup (i.e., pre-train on SleepEEG and fine-tune on Epilepsy), we evaluated clustering on learned representations $z_i^{\textrm{tune}}$ in the Epilepsy test set. Specifically, we used K-means clustering with K=2, as Epilepsy has 2 classes, on top of $z_i^{\textrm{tune}}$ in fine-tuning. We adopt commonly used evaluation metrics: Silhouette score, ARI, and NMI. The following table shows that TF-C obtains the best clustering performance and surpasses the strongest baseline (TS-TCC) by a large margin (5.4% in the Silhouette score). However, we also observe that TNC and CPC performed worse than TS-TCC and TF-C. As discussed earlier, TNC is intrinsically more advantageous on longer time series samples, and CPC relies on relatively stationary samples. This explains why these baselines struggle with clustering when asked to transfer the latent space from SleepEEG to Epilepsy.
>
> |Method | Silhouette |ARI |NMI|
> |-------|---------------|----------|--------|
> Non-DL (KNN) |  0.1208 +/-0.0271	|0.0549 +/- 0.0059	|0.0096+/- 0.0014|
> Random Init. | 0.3497 +/- 0.0509 |0.3216 +/-0.0136  |0.2408+/- 0.0287 |
> TNC | 0.2353 +/- 0.0018 | 0.0211+/- 0.0061| 0.0082+/- 0.0018|
> CPC| 0.2223 +/-0.0011| 0.0153 +/- 0.0196| 0.0063 +/- 0.0055|
> TS-TCC | 0.5154 +/- 0.0458| 0.6307 +/- 0.0325| 0.5178 +/- 0.0283|
> TF-C |0.5439 +/- 0.0417 |0.6583 +/-0.0259 | 0.5567+/-0.0172 |
>
> - **Anomaly detection:** We added new experiments to evaluate how TF-C performs for anomaly detection. Note we work on the sample- rather than observation-level anomaly detection. Based on global patterns, the former aims to detect abnormal time series samples instead of outlier observations in the samples (as used in BTSF [R1] and USAD [R2]), which emphasizes the local context. In one-to-one setup FD-A $\rightarrow$ FD-B, we built a small subset of FD-B with 1000 samples, of which 900 are from undamaged bearings, and the other 100 are from bearings with inner or outer damage. Undamaged samples are considered “normal,” and inner/outer damaged samples are “outliers.” In fine-tuning, we used one-class SVM on top of learned representations $z_i^{\textrm{tune}}$. Details about the dataset and setup will be added to the camera-ready version. The experiments are run once (no std) for efficiency reasons. We report the F-1 score as AUROC and AUPRC are not applicable (one-class SVM doesn’t generate the prediction probabilities). The following table shows results for TF-C and five competitive baselines. It illustrates that our TF-C is sensitive to anomalous samples and applicable to real-world monitoring of mechanical states.
>
> |Methods | Non-DL(KNN) | Random Init. | TNC | CPC| TS-TCC | TF-C |
> |-------|---------------|----------|--------|--------|------------|---------------|
> F-1 score | 0.5061 | 0.7639 |0.7957 | 0.5238 | 0.4759 | 0.8312|
>
>
> (3) Finally, we agree that forecasting is a promising future direction. We will discuss it in the subsection of limitations and future work in the camera-ready version. For example, our TF-C principle can be integrated with the structure of Informer [R3] for long sequence time-series forecasting.
>
> [R2] Audibert J, Michiardi P, Guyard F, et al. USAD: Unsupervised anomaly detection on multivariate time series, KDD. 2020: 3395-3404.
>
> [R3] Zhou H, Zhang S, Peng J, et al. Informer: Beyond efficient transformer for long sequence time-series forecasting, AAAI. 2021, 35(12): 11106-11115.

---

> ### Author Response · Authors · 2022-08-02
> **Response to Reviewer hNJe Part IV**
>
> **Q4**: Further understanding of frequency augmentation is needed. This paper verified its effectiveness but didn't provide further analysis about how it works. How Are different augmentations compared in the frequency domain?
>
> **A4**: Thanks for the thoughtful comment. We have thoroughly explored various frequency augmentations. We are sorry the paper did not discuss all of them, and we will work on clarifying them. To address this, we here analyze eight frequency augmentation policies and discuss their performance.
>
> We consider three types of frequency augmentations:
>
> - *Low- vs. High-band perturbations:* High-frequency components contribute to the fast varying parts of the time series (e.g., sharp transitions). In contrast, low-frequency components contribute to slow signal variations in the time domain. Low band perturbations correspond to the first half of the frequency spectrum.
>
> - *Single- vs. Multi-component perturbations (aka varying budget size):* Perturbing several frequency components (i.e., larger budget) leads to more substantial changes in the time domain. In this analysis, we set the budget to 5 for multi-component perturbations and 1 for single-component perturbations.
>
> - *Random vs. Distributional perturbations:* The selection of components in the spectrum that get perturbed is important because it influences the resulting time-based representation of the augmented sample. Frequency components can be sampled uniformly at random or according to a user-specified distribution (i.e., Gaussian in the following analysis).
>
> Based on the above considerations, we developed eight frequency augmentations. To visualize perturbations in the frequency domain and their effects in the time domain (through the inverse FFT), we include a new figure in the paper (Fig. 6 with ten panels; Epilepsy dataset; see revised Appendix).
>
> In addition, we conducted new experiments to explore how different spectral perturbations affect TF-C performance. The following table shows results for the one-to-one setup (SleepEEG $\rightarrow$ Epilepsy). We find the following:
>
> - Perturbing low-band components performs better than augmenting high-band elements of the spectrum (0.8% change in F1 performance on average across the four setups). This can be explained by the nature of the underlying time series—in physiological signals, most information is carried in a low band (e.g., EEG signals are most informative in 0.5-70 Hz while > 70Hz is considered noise). This means that the choice of low- vs. high-band components depends on the signal context. For example, high-band components are informative for mechanical signals. We further verified this by external experiments in the FD-A $\rightarrow$ FD-B setting, where perturbing high-band components outperform low-band augmentations by 0.8% in F1. Thus, we consider perturbations across the entire spectral band in the paper to achieve broad generalization.
>
> - In general, perturbing many frequency components degrades model performance. One reason for worse performance is that too big a budget incurs too significant changes in the time-based representation of augmented samples. In that case, the augmented samples would be easily distinguished by a contrastive model, leading to poor contrastive encoders. For this reason, we use single-component perturbations in the paper.
>
> - We cannot draw a clear conclusion on whether the random selection of components is better than Gaussian distribution-based as both F1 scores are very similar. The optimal choice of sampling strategy likely depends on both the time-series signal and prediction task. We use random sampling in the paper.
>
> We will include the above analyses, experimental results, and insights in the camera-ready version. In addition, we note that the developed augmentations in this paper perturb magnitude and keep phase information unchanged. The phase of perturbation is a crucial direction in frequency domain augmentation in future work. Finally, we agree that frequency augmentations represent a fruitful direction for future research, which we will address in the limitation and future work section.
>
>
> | Frequency augmentations | Low vs. High | Single vs. Multi | Random vs. Gaussian | F1 |
> |-------|--------------|--------------|--------------|--------------|
> | 1| Low  | Single | Random | 0.9136 |
> | 2| Low  | Single | Gaussian| 0.8957 |
> | 3| Low  | Multi | Random | 0.8763 |
> | 4| Low  |  Multi | Gaussian| 0.8859|
> | 5| High | Single | Random | 0.8972 |
> | 6| High  | Single | Gaussian| 0.8651 |
> | 7| High  |  Multi | Random | 0.8818 |
> | 8| High  |  Multi | Gaussian| 0.8775 |

---

### Official Review · Reviewer_q9DY · 2022-07-13

**Rating:** 6
**Confidence:** 5
**Soundness:** 3 good
**Presentation:** 4 excellent
**Contribution:** 3 good

**Summary:**

In this work, the authors present a self-supervised pre-training model for time series (TS) data by leveraging Time-Frequency Consistency (TF-C). The main idea is that time-based and frequency-based representations of the same time series sample are closer in the joint time-frequency space, and farther apart from the representations of other samples. The model has 4 components: a time encoder - to embed TS from time domain; a frequency encoder - to embed TS from frequency domain; and two cross-space projectors that map time-based and frequency-based representations to a unified time-frequency space. The model is trained using 3 types of losses: Contrastive time loss, Contrastive frequency loss, and Time-frequency consistency loss. To enable contrastive learning, the model uses augmentation banks for both time and frequency embeddings. The authors evaluate their method in two experimental scenarios, (a) pre-train on one dataset and test on one dataset (b) pre-train on one dataset and test on multiple datasets with 8 diverse datasets and compare it against 8 baselines methods to show the superiority of TF-C approach.

**Questions:**

1. The current embedding scheme and objective functions focus on global time series features which are suitable for the classification task. Can the proposed method be applied easily to time series forecasting task where modeling local features is relevant as well?

2. Did you try pre-training the model on multiple datasets as well?



**Limitations:**

See weaknesses.

**Strengths And Weaknesses:**

Strengths:

1. The paper is well written and easy to follow. I enjoyed reading it.
2. While frequency domain has been modeled in recent methods like CoST, the idea of using Time-Frequency Consistency, in addition to individual domain losses, for self-supervision is novel.
3. The method was compared to 8 baselines methods on 8 diverse datasets and showed strong empirical results. Through an ablation study of loss function components, the paper confirms that each loss term contribute towards the model performance.
4. The paper performed multiple experimental settings (one-to-one and one-to-multiple) and dataset scenarios (transfer learning among datasets with substantial discrepancy) to show the method outperforms the other baselines.

Weaknesses:
1. The authors present the proposed method as universal pre-training method for time-series, however, only a single time series task (time-series classification) is explored in the paper. Therefore, it is not clear if the proposed method performs competitively on other time series tasks.

Typos:

(a) Line 173 - shouldn’t it be D^tune instead if D^pret?
(b) Line 175 -  “D^pret and D^pret” ->  “D^pret and D^tune”
(c) Line 221 - “sale” ->   scale

---

> ### Author Response · Authors · 2022-08-02
> **Response to Reviewer q9DY Part I**
>
> Thank you for your helpful feedback! We appreciate that you felt our method is reasonable and that you recognize the results are impressive and valuable for many applications. Below, we briefly respond to each of your concerns. If, after reading our responses, you are still not convinced of the merits of our work, we would love to know what else we can do to help improve the work towards a better score in your eyes.
>
> **Q1**: The current embedding and objective functions focus on global time series features suitable for classification. Can the TF-C method be applied to time series forecasting tasks where local features are also relevant?
>
> **A1**: Thank you for this valuable comment. Our response has three parts:
>
> (1) First,  while we focus on knowledge transfer across diverse time series datasets, we agree that the current embedding scheme and loss functions are more suitable to capture global representations of time series. Our model was designed to capture the generalizable pattern aiming at knowledge transfer, which is favorable for classification tasks that leverage global information over tasks that use local context (e.g., forecasting). As you suggested in the review, in the camera-ready version, we will update our manuscript thoroughly (e.g., title, abstract, introduction, methods) to clarify that this work is designed mainly for time series classification.
>
> (2) Second, beyond the above point, we did new experiments to demonstrate how our approach applies to clustering and anomaly detection. We compared our TF-C with 5 baselines on these tasks: 2 non-transfer-based baselines (Random Init. and Non-DL), the best performing baseline currently included in the paper (i.e., TS-TCC), and 2 new methods that we included just for this experiment (TNC and CPC, suggested by Reviewer hNJe).
>
> - **Clustering:** In one-to-one (i.e., SleepEEG $\rightarrow$ Epilepsy), we evaluated clustering on learned representations $z_i^{\textrm{tune}}$ in the Epilepsy test set. Specifically, we added a K-means clustering (K=2; Epilepsy has 2 classes), on top of $z_i^{\textrm{tune}}$ in fine-tuning. We adopt commonly used evaluation metrics: Silhouette score, ARI, and MI. The following table shows our TF-C obtains the best clustering surpassing the strongest baseline (TS-TCC) by a large margin (5.4% in Silhouette score). It conveys that TF-C can capture more distinctive representations with the knowledge transferred from pre-training, which is consistent with the superiority of TF-C in the classification task.
>
> |Method | Silhouette |ARI |NMI|
> |-------|---------------|----------|--------|
> Non-DL(KNN) |  0.1208 +/-0.0271	|0.0549 +/- 0.0059	|0.0096+/- 0.0014|
> Random Init. | 0.3497 +/- 0.0509 |0.3216 +/-0.0136  |0.2408+/- 0.0287 |
> TNC | 0.2353 +/- 0.0018 | 0.0211+/- 0.0061| 0.0082+/- 0.0018|
> CPC| 0.2223 +/-0.0011| 0.0153 +/- 0.0196| 0.0063 +/- 0.0055|
> TS-TCC | 0.5154 +/- 0.0458| 0.6307 +/- 0.0325| 0.5178 +/- 0.0283|
> TF-C |0.5439 +/- 0.0417 |0.6583 +/-0.0259 | 0.5567+/-0.0172 |
>
> - **Anomaly detection:** We added new experiments to evaluate TF-C on anomaly detection. Note we work on the sample- rather than observation-level anomaly detection. Based on global patterns, the former aims to detect abnormal time series samples instead of outlier observations in the samples (as BTSF [R1] and USAD [R2]), which emphasizes the local context. In setup FD-A $\rightarrow$ FD-B, we built a small subset of FD-B with 1000 samples, of which 900 are from undamaged bearings, and the other 100 are from bearings with inner or outer damage. Undamaged samples are considered “normal,” and inner/outer damaged samples are “outliers.” In fine-tuning, we used one-class SVM on top of learned representations $z_i^{\textrm{tune}}$. Details about the dataset and setup will be added to the camera-ready version. The experiments are run once (no std) for efficiency reasons. The following table shows results for TF-C and five competitive baselines. It illustrates that our TF-C is sensitive to anomalous samples and applicable to real-world monitoring of mechanical states.
>
> |Models | Non-DL(KNN) | Random Init. | TNC | CPC| TS-TCC | TF-C |
> |-------|---------------|----------|--------|--------|------------|---------------|
> F-1 Score | 0.5061 | 0.7639 |0.7957 | 0.5238 | 0.4759 | 0.8312|
>
> (3) Finally, we agree that forecasting is a promising future direction. We will discuss it in the subsection of limitations and future work in the camera-ready version. For example, our TF-C principle can be integrated with the structure of Informer [R3] for long sequence time-series forecasting.
>
> [R1] Yang L, Hong S. Unsupervised Time-Series Representation Learning with Iterative Bilinear Temporal-Spectral Fusion. ICML, 2022.
>
> [R2] Audibert J, Michiardi P, Guyard F, et al. USAD: Unsupervised anomaly detection on multivariate time series, KDD, 2020.
>
> [R3] Zhou H, Zhang S, Peng J, et al. Informer: Beyond efficient transformer for long sequence time-series forecasting, AAAI, 2021.

---

> ### Author Response · Authors · 2022-08-02
> **Response to Reviewer q9DY Part II**
>
> **Q2**: Did you also try pre-training the model on multiple datasets?
>
> **A2**: Thank you for this insightful suggestion about a setup to pre-train the model on multiple datasets (i.e., many-to-one setting). Our response has two parts.
>
> (1) First, the many-to-one setup fundamentally differs from the one-to-one and also from one-to-many configurations studied in our initial submission.
>
> - Notably, in the one-to-one and one-to-many setups, the model is pre-trained on a single pre-training dataset presumed to originate from a data source exhibiting homogeneity. For example, samples in the pre-training dataset could describe similar devices (e.g., different kinds of boring machines) with similar sampling rates, lengths, semantic meanings, and/or relevant temporal patterns that occur on comparable time scales, etc. When some level of homogeneity exists in a pre-training dataset, the method can learn patterns and apply them to different fine-tuning datasets.
>
> - In contrast, in many-to-one setups, pre-training datasets can be incredibly heterogeneous. For example, a pre-training dataset might contain samples describing boring machines' behavior and recordings of EEG medical devices. For this reason, pre-training can be severely hampered by diversity and misalignments across samples in the pre-training dataset. When we merge many heterogeneous samples into a single pre-training dataset, the model might struggle to find patterns underlying the combined dataset. While models in computer vision have been successfully pre-trained on large, diverse datasets, such as ImageNet, the literature on pre-training for time series is much sparser. Further, heterogeneous pre-training datasets pose additional challenges due to the unique properties of time series. Learning universal representations from multi-sourced time series datasets (i.e., many-to-one setups) remains an open challenge. It is an interesting future direction, and we hope this study can facilitate future research.
>
>
> (2) In response to your question, we did four new experiments for the many-to-one setups, named N-to-one for N = 1,2, 3, and 4. The descriptions and results of our experiments are explained below.
>
> - Take four-to-one as an example. We merge four pre-training datasets (SleepEEG, FD-A, HAR, and ECG) into one pre-training dataset. We had to preprocess the datasets to make a many-to-one setup feasible. To address the mismatch between the number of channels, for the multivariate dataset (HAR), we take a single channel (i.e., x-axis acceleration) to make sure all samples have the same number of channels. To align the length across datasets, we make all samples have 1500 observations by zero-padding (the ones shorter than 1500) and cutting-off (the ones longer than 1500). Further, we take 25% samples (instead of 100%; for computational reasons only) from each of the four datasets to create a pre-training dataset with 108,300 samples that belong to 18 classes in total. Similarly, we built pre-training for three-to-one (merge SleepEEG, FD-A, and HAR) and two-to-one (merge SleepEEG and FD-A).
>
> - We take the above N-to-one pre-training datasets and repeat the following analysis. First, we pr-e-train our TF-C and then fine-tune it on Epilepsy for classification. The table below shows that AUROC decreases with the increasing heterogeneity of pre-training datasets, which is consistent with our analysis in the first part of this answer. Nevertheless, whether there exists a framework that overcomes the problems and enables many-to-one pre-training for time series remains to be seen, and we think it's an exciting direction for future research.
>
> | Setup | AUROC|
> |------|-------------|
> |One-to-one| 0.9819|
> |Two-to-one| 0.9525|
> |Three-to-one| 0.8607|
> |Four-to-one| 0.7253|

---

### Author Response · Authors · 2022-08-02
**General response**

Thank you to all reviewers for the thoughtful feedback.

We are pleased that all four reviewers agree that the paper is clearly written and easy to follow. We are also delighted that reviewers recognized this study as introducing "a novel yet simple solution to an important problem" that advances "a less explored area of self-supervised learning with time series." Further, we are grateful for acknowledging the "remarkable results against other current baselines" and "comprehensive experimental settings and ablation studies."

In response to reviewers' comments, we performed 5 new experiments and included 2 new baselines. In addition, we generated 6 new tables and 1 figure summarizing these additional studies. Together with various minor revisions throughout the manuscript, we hope these updates address all key concerns and clarify the significance of our work.

We respond to all your comments below in the individual replies, where we directly address comments raised by individual reviewers. Please note that in our response, all references prefixed with "R" (e.g., [R1]) are newly added citations in this rebuttal. References without the prefix "R" are citations used in our original submission. We uploaded the revised manuscript (updates are colored in blue) and expanded the Appendix to Open Review for everyone to see. In addition, we carefully proofread the paper, fixed a bunch of typos, and edited the paper for clarity. We will include further details in the camera-ready version (revisions during the rebuttal are still limited to 9 pages of the main paper).

Thank you again for your thoughtful commentary. We worked hard to improve our paper, and we sincerely hope you will find our responses informative and helpful.

If you feel we have not sufficiently addressed your concerns to motivate increasing your score, we would love to hear from you further on what points of concern remain and how we can improve the work in your eyes. Thank you again!

---

### Author Response · Authors · 2022-08-06
**Looking forward to Reviewers’ Feedback**

Dear Reviewers,

We have responded in detail to the concerns of all reviewers, added new experiments and case studies, added external baselines, and revised statements along with various minor revisions to emphasize our significance and clarify existing confusions. After posting our responses, we would be happy to hear back from the reviewers (especially Reviewer hNJe) that: do our responses answer the questions, and what we can do to further increase our score?  Even if reviewers are confident in the initial assessments, we would love to receive additional clarification on your feedback.

Thank you very much for your time!

---

> ### Author Response · Authors · 2022-08-08
> **Thanks for updating your comments**
>
> We noticed some reviewers have updated their comments. We’d like to express our gratitude for reading and considering our responses.
>
> Thank you for your time again!

---

### Meta-Review · Area_Chair_vZh8 · 2022-08-27

**Recommendation:** Accept
**Confidence:** Certain

**Metareview:**

The reviewers highlight the novelty and significance of the proposed approach, the extensive empirical evaluation, and the clarity of writing. Additional experimental results provided during the rebuttal cleared up some concerns around the general applicability of the approach.

**Award:**

No

---

### Decision · Program_Chairs · 2022-09-14

Accept